# Anisotropic organization of circumferential actomyosin characterizes hematopoietic stem cells emergence in the zebrafish

Mylene Lancino[1,2,3], Sara Majello[1,2], Sebastien Herbert[1,2,4], Fabrice De Chaumont[5,6], Jean-Yves Tinevez[4], Jean-Christophe Olivo-Marin[5,6], Philippe Herbomel[1,2]*, Anne Schmidt[1,2]*

[1]Department of Developmental and Stem Cell Biology, Institut Pasteur, Paris, France; [2]CNRS, UMR 3738, Paris, France; [3]Sorbonne Université, UPMC Paris 06, Complexité du Vivant, Paris, France; [4]Image Analysis Hub, UTechSPhotonic BioImaging (Imagopole), Citech, Institut Pasteur, Paris, France; [5]Department of Cell Biology and Infection, Institut Pasteur, Paris, France; [6]CNRS, UMR3691, Paris, France

**Abstract** Hematopoiesis leads to the formation of blood and immune cells. Hematopoietic stem cells emerge during development, from vascular components, via a process called the endothelial-to-hematopoietic transition (EHT). Here, we reveal essential biomechanical features of the EHT, using the zebrafish embryo imaged at unprecedented spatio-temporal resolution and an algorithm to unwrap the aorta into 2D-cartography. We show that the transition involves anisotropic contraction along the antero-posterior axis, with heterogenous organization of contractile circumferential actomyosin. The biomechanics of the contraction is oscillatory, with unusually long periods in comparison to other apical constriction mechanisms described so far in morphogenesis, and is supported by the anisotropic reinforcement of junctional contacts. Finally, we show that abrogation of blood flow impairs the actin cytoskeleton, the morphodynamics of EHT cells, and the orientation of the emergence. Overall, our results underline the peculiarities of the EHT biomechanics and the influence of the mechanical forces exerted by blood flow.
DOI: https://doi.org/10.7554/eLife.37355.001

*For correspondence:
philippe.herbomel@pasteur.fr (PH);
anne.schmidt@pasteur.fr (AS)

**Competing interests:** The authors declare that no competing interests exist.

## Introduction

Hematopoiesis is an essential biological process leading to the genesis of all blood lineages throughout life. Currently, producing full-potential hematopoietic stem cells (HSCs) in vitro remains a challenge (*Yvernogeau et al., 2016*), in particular for regenerative purposes. This points to the relevance of reinforcing the fundamental knowledge of the process as it occurs in vivo.

HSCs with long-term replenishment potential are formed in the embryo according to a genetic program highly conserved between vertebrate species (*Ciau-Uitz et al., 2014*; *Ciau-Uitz and Patient, 2016*; *Jaffredo and Yvernogeau, 2014*). HSCs emerge during the so-called definitive wave of hematopoiesis, from vascular components, in particular the ventral floor of the dorsal aorta, in the aorta-gonad mesonephros (AGM) region as well as from the umbilical and vitelline arteries in mammals.

The zebrafish has proven to be an extremely valuable model for studying the developmental aspects of hematopoiesis (*Frame et al., 2017*); the combination of cell tracing and live-imaging have notably allowed to trace the migration of definitive hematopoietic stem and progenitor cells (HSPCs)

**eLife digest** As humans, we have two major types of blood cell: our red blood cells transport oxygen around the body, while our white blood cells fight disease. Both types of cell come from the same stem cells, which first appear early in embryonic development. These stem cells emerge from the walls of major blood vessels, including the aorta – which carries blood from the heart.

Stem cells have not yet decided which adult cell to become. Given the right signals, blood stem cells can form red blood cells or any of the different types of white blood cell. Understanding this process could allow scientists to recreate it in the laboratory, making blood stem cells that can give rise to all blood cells found in the body. But, this is not yet possible because we do not know all the conditions needed to make the cells and ensure they survive. One crucial gap in our understanding concerns the importance of blood flow. As the main blood vessel leaving the heart, the aorta experiences mechanical stress every time the heart beats. Lancino et al. wanted to find out whether this influences the development of the blood stem cells.

Zebrafish embryos are transparent, making it easy to see their bodies developing under a microscope. Like humans, they also produce both red blood cells and white blood cells meaning Lancino et al. could watch the birth of blood stem cells in these embryos from a part of the aorta called the aortic floor. A new software tool unwrapped pictures of the tube-shaped blood vessel into flat, two-dimensional maps, making it possible to see how the aorta changed over time. This revealed that, as blood stem cells leave the aortic floor, they bend and contract with the direction of the blood flow. Rings of actin and myosin proteins that formed around the stem cells as they are born helped the process along, while stopping the heartbeat changed the way the blood cells emerged. Without any blood flow, the actin proteins did not assemble properly; the stem cells also emerged in the wrong direction and some of them even died.

These findings show that physical forces play a role in the formation of blood stem cells. Understanding this process brings scientists a step closer to recreating the conditions for making different kinds of blood cells outside of the body.

DOI: https://doi.org/10.7554/eLife.37355.002

to successive hematopoietic organs (*Jin et al., 2007*; *Kissa et al., 2008*; *Murayama et al., 2006*). Real-time in vivo imaging also established unambiguously the aortic origin of definitive HSPCs, that were visualized emerging as single cells directly from the hemogenic endothelium in the floor of the dorsal aorta (*Bertrand et al., 2010*; *Kissa and Herbomel, 2010*). This very unique type of cell emergence, called the endothelial-to-hematopoietic transition (EHT), is characterized by an unusual bending of the cell that increases while the emergence is progressing, and so until the exit from the vascular wall (*Kissa and Herbomel, 2010*). The process was also described ex vivo in slices of mouse embryos, although at a lower resolution (*Boisset et al., 2010*). An apparent difference between zebrafish and mouse is the direction of the emergence, with HSPCs extruding into the sub-aortic space in the former and into the aortic lumen in the latter (where they initiate the so-called intra-aortic clusters). The reasons for such a difference are currently unknown and may relate to different biophysical constraints of the aortic and sub-aortic environments in either species.

The vascular and aortic systems have very unique characteristics (*Aird, 2012*). They are constituted by extremely flat endothelial cells with luminal/apical and basal membranes having virtually equal surfaces (a so-called squamous epithelium) and organized around a lumen filled with blood cells and fluid, and exposed to high mechanical loading, including shear stress exerted on the surface of endothelial cells as well as circumferential wall stress (*Lu and Kassab, 2011*). Blood flow also appears to regulate endothelial cell fates in development (*García-Cardeña and Slegtenhorst, 2016*) and, in regard to hematopoiesis, to have an influence on the expression of key transcription factors such as Runx1, the transcription factor essential for hematopoiesis, both in mouse embryonic stem cells (*Adamo et al., 2009*) and in the zebrafish (*North et al., 2009*). Hence, comprehending the fundamental mechanisms of HSC emergence requires taking into account the peculiarities of the vascular environment and the mechanical constraints.

The EHT allows single live cells to leave a squamous epithelium without perturbing its integrity. Most other cell extrusion processes that have been described concern thick, columnar/cuboidal

epithelia, and among them, only the epithelial mesenchymal transition (EMT, [*Lamouille et al., 2014*; *Nieto et al., 2016*]) gives rise to live wandering cells endowed with a new potential, as the EHT does. Hence, in order to specify further the peculiarities of the EHT and to be able to establish comparisons with the other cell extrusion processes described so far, we delved deeper into the process, taking the vascular environment into account. To do so, we developed a novel algorithm allowing the deployment of the aortic wall into high-resolution 2D-cartographies and 2D-time-lapse sequences. We also addressed the influence of blood flow on the EHT morphodynamics by abrogating heart beating. This led us to uncover some of the key and unique features of the EHT (and to draw a comprehensive model of its main steps) and identify the contribution of some of the essential molecular components involved in the morphological changes accompanying the emergence. Further, our analysis brought insight into the dynamic interplay between cells undergoing EHT and adjoining endothelial cells and pointed to the essential role of blood flow in HSCs emergence.

## Results

### EHT dynamics at high spatio-temporal resolution

In order to unravel the critical biomechanical features of the EHT, we performed confocal live-imaging at high spatio-temporal resolution. In comparison to previous work (*Kissa and Herbomel, 2010*), we improved the spatial resolution of our time-lapse (TL) sequences by reducing the distance between the sequential optical planes (0.43 μm instead of 2 or 0.60 μm) and improved the temporal resolution by reducing the 5 min intervals between Z-stacks acquisitions down to 2 min when performing spinning-disk confocal imaging. We imaged the dorsal aorta (DA) of transgenic zebrafish embryos (*Figure 1A*), starting at 48 hpf (corresponding to the timing at which the EHT process is culminating, see *Kissa and Herbomel, 2010*), until 72 hpf. We noticed that the angles of emergence of EHT undergoing cells (hereafter designated as 'EHT cells') varied between approximately 0–45° from the dorso-ventral axis of the DA and 0–45° from its antero-posterior (A-P) axis. In general, the most valuable information was obtained from cells for which the two most distant poles were parallel to the A-P axis (*Figure 1B*). We initiated the study using *Tg(kdrl:Ras-mCherry; kdrl:eGFP)* fish so as to visualize cellular membranes as well as the cytoplasmic volume. As previously described (*Kissa and Herbomel, 2010*), the morphological criterion allowing unambiguous identification of cells having initiated the EHT is their cup-shaped morphology, with bending toward the sub-aortic space. Hence, many of our TL sequences were initiated at this stage, increasing chances to image completion of the process and minimizing the risk of phototoxicity (see *Figure 1C* for a 3D-rendering view, and *Figure 1—video 1*, *Figure 1—video 2*). Ras-mCherry allowed visualizing the luminal and basal membranes (*Figure 1H*), revealing that the latter underwent more or less extensive blebbing at the cup-shaped stage (*Figure 1D,I*). This blebbing preceded the protrusion of large membrane extensions that were formed hours before the cell exit and were reminiscent of cell shape changes occuring during amoeboid migration (*Figure 1—video 1*). Finally, at the end of the process, Ras-mCherry delineated a transient narrow membrane foot that remained connected to the aorta floor and preceded release in the sub-aortic space (*Figure 1F,G and L* and *Figure 1—video 1* and *Figure 1—video 2*).

We then measured the time-to-exit, starting from the onset of imaging at the cup-shaped stage and found that it varied widely, with an average duration of 5.48 hr (n = 66, see *Figure 1—figure supplement 1B*, right panel (note that we define the cup-shaped stage as the steps at which the angle between the axis tangential to the inner cell border and the plane of the aortic floor evolves from 150 to 90°, see A: middle top panel)). We also imaged the initial phase of the cell bending, starting from a flat-shaped to the cup-shaped stage in order to determine the time required to reach the latter. We followed a limited amount of cells because, in addition to increasing the risk of phototoxicity owing to long hours of laser exposure, it is currently impossible to determine the precise starting point at which EHT committed cells, engulfed in the hemogenic endothelium, will initiate the emergence (hence hampering the precise determination of EHT initiation and duration). The time intervals between the flat morphology to the cup-shaped stage were also very variable, with an average of 6.5 hr (n = 7, see *Figure 1—figure supplement 1B*, left panel and *Figure 1—video 3*).

Finally, to quantify the progression of the process, starting from the cup-shaped stage, we manually measured through time the distance between the anterior and posterior poles of the EHT cell

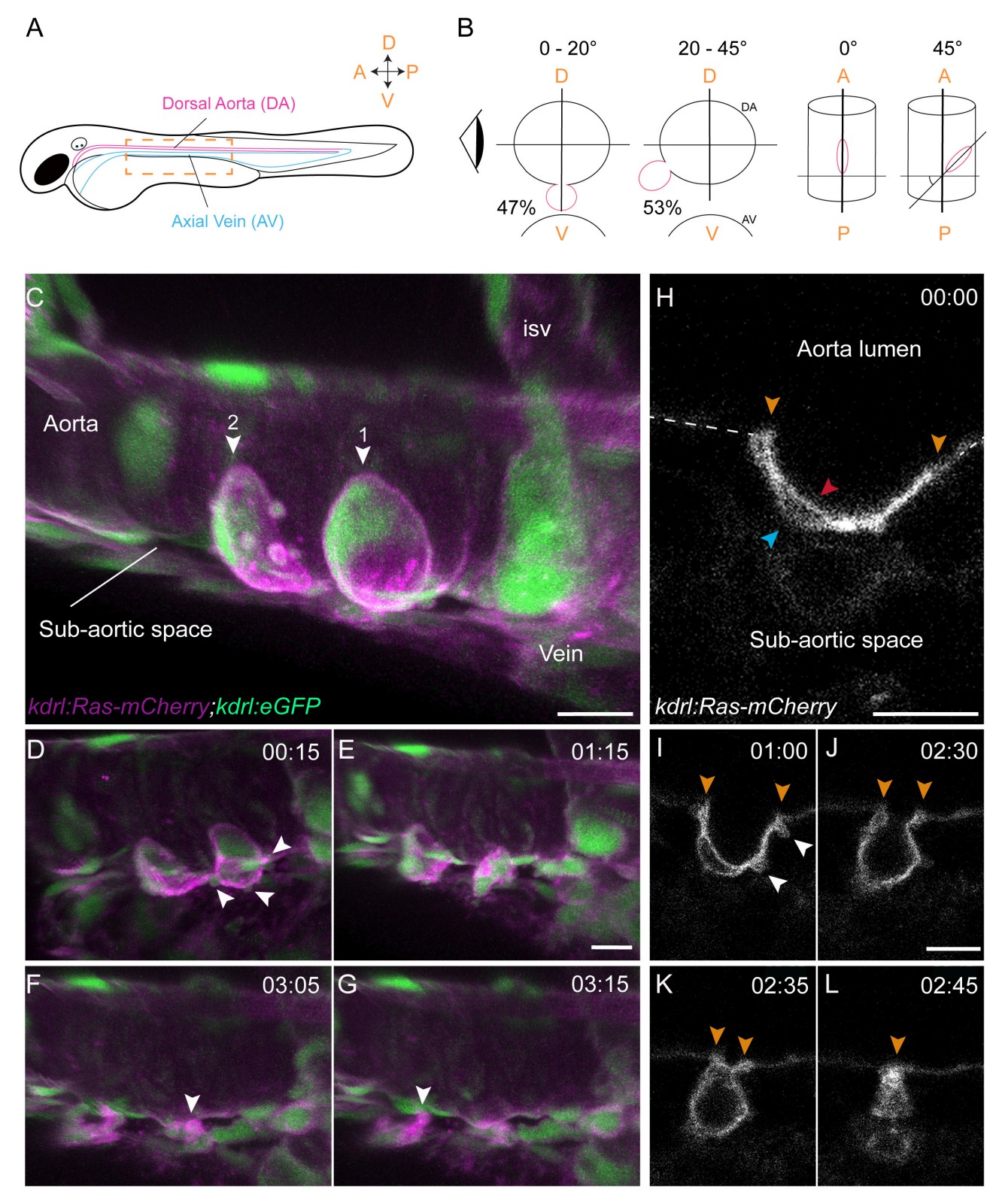

**Figure 1.** Sequential steps and morphological changes during the EHT (A–B) The EHT is variable in space and time. Schematic representations of (A) a zebrafish embryo at 48 hpf; a yellow rectangle shows the region of imaging. (B) Left, transversal sections of the dorsal aorta showing the % of cells undergoing emergence (in red) at 0°−20° or 20°–45° angle relative to the dorso-ventral axis (N = 49 cells). The eye looks in the direction of imaging. Right, top view showing variation of the angle of emergence (with the A-P axis as reference). Note that the EHT is also characterized by variability in its
*Figure 1 continued on next page*

*Figure 1 continued*

time-length, see *Figure 1—figure supplement 1* and main text. (**C–L**) Live confocal images from 48 hpf *Tg(kdrl:Ras-mCherry; kdrl:eGFP)* embryos. (**C–G**) Images extracted from a 3D-rendering TL sequence (**D–G**) and a Z-stack acquired 120 min before initiation of the time-lapse (**C**), showing the typical changes of cell shape during the EHT (see *Figure 1—video 1*). (**C**) Numbered arrowheads: rim of two cup-shaped EHT undergoing cells. Arrowheads indicate blebs in (**D**) and cellular foots in (**F and G**). isv: intersegmental vessel (see also *Figure 1—video 1*). (**H–L**) Single Z-planes corresponding to cell #2 extracted from the same TL sequence. Arrowheads: cell borders connecting with adjoining endothelial cells (in yellow), the luminal membrane (in red), the basal membrane (in blue), and blebs (in white), respectively (see *Figure 1—video 2*). Time is indicated in hrs:min. Scale bars, 10 μm.

DOI: https://doi.org/10.7554/eLife.37355.003

The following video and figure supplement are available for figure 1:

**Figure supplement 1.** The time-length of the EHT is very heterogeneous (see text also).

DOI: https://doi.org/10.7554/eLife.37355.004

**Figure 1—video 1.** EHT at high spatio-temporal resolution.

DOI: https://doi.org/10.7554/eLife.37355.005

**Figure 1—video 2.** Luminal/apical membrane invagination of an EHT undergoing cell.

DOI: https://doi.org/10.7554/eLife.37355.006

**Figure 1—video 3.** The EHT- from the flat hemogenic endothelium to the emergence in the sub-aortic space.

DOI: https://doi.org/10.7554/eLife.37355.007

(within the plane of the aortic wall), which we will refer to as the A-P luminal distance (*Figure 1—figure supplement 1C*). Firstly, we found little correlation between the A-P luminal distance at the beginning of our imaging and the time of exit of the EHT cell (the longest A-P luminal distances did not imply the longest times-to-exit, see *Figure 1—figure supplement 1D*). Secondly, we noticed that the narrowing down of the A-P luminal distance did not occur at a uniform pace, but with phases of contraction and of relative stabilization of variable durations taking place in the limit of an A-P luminal distance of approximately 2 μm, followed by steep decrease until the release of the cell (this fast drop may represent a collapse following the culmination of mechanical tensions). Hence, the heterogeneous duration of the EHT might be explained by these discontinuities, but the reason for this is unknown.

## Intercellular junctions and apical constriction of EHT cells are anisotropic

A specific feature of the EHT is that EHT cells maintain contact with their endothelial neighbors until the very end of the process. This is most probably because of the need to maintain the tightness of the endothelium. In order to visualize the local density and spatial organization of junctional complexes established between EHT cells and contacting endothelial neighbors, we followed ZO1, a protein that interacts preferentially with tetraspan membrane constituents of tight junctions, and also with α-catenin (hence with cadherin complexes, [*Maiers et al., 2013*]). We performed live-imaging of hemogenic regions of the aortic floor using *Tg(fli1:Gal4; UAS:RFP; UAS:eGFP-ZO1)* (*Herwig et al., 2011*) embryos (with cytosolic RFP facilitating the localization of cells in the ventral side of the aorta whose increase in thickness indicates hemogenic potential). We observed a heterogeneous density of eGFP-ZO1 at the interfaces between EHT cells and their endothelial neighbors, with an apparent increase at the anterior and posterior poles of the cell that persisted until their convergence to apparent coalescence, shortly before cell exit (*Figure 2A* and *Figure 2—video 1*).

We then tracked semi-automatically the displacement of these eGFP-ZO1 enriched regions at the poles and used them as reference to follow the apical constriction as they converge concomitant to the closure of the circumferential junctional belt. This complemented, with a higher spatio-temporal resolution, our results presented in *Figure 1—figure supplement 1D* on the variability of the temporal course of the EHT process per se and led to two main conclusions. Firstly, the spots of eGFP-ZO1 high density at the two opposite poles of the EHT cell moved mainly along the A-P axis, often with an imbalance of migration distance (*Figure 2B* and legend). Secondly and importantly, the distance between the two eGFP-ZO1 high-density regions decreased non-uniformly (*Figure 2C* and *Figure 2—figure supplement 1*), with phases of contraction and relative stabilization (already visible in *Figure 1—figure supplement 1D* and better resolved spatio-temporally here owing to semi-automatic tracking, see Materials and methods). The successive phases of contraction and stabilization are characterized by oscillations of the closing speed of relatively homogenous periodicity

(*Figure 2C,D*), each oscillation resulting from acceleration and deceleration phases. The oscillations taking place during the contraction phases are the largest in amplitude and are interspersed by short periods of time during which the progression of the A-P distance reaches a minimum (referred to as pseudo-plateaus, see *Figure 2—figure supplement 1*). This oscillatory behavior of the apical constriction is reminiscent of the pulsatile/oscillatory behavior observed in several morphogenetic events during development (*Levayer and Lecuit, 2012*), although with significant differences in the biomechanical parameters of these oscillations (see Discussion).

We then thought of investigating more systematically the localization of eGFP-ZO1 in junctional regions across the entire aortic wall, as well as using the delineation of cell contours by this reporter to study in detail the EHT cellular landscape, notably the morphology of EHT cells and adjoining endothelial ones. Precise morphological information on the aortic cellular landscape is difficult to extract from Z projections or 3D views, and projections mask structures in the medial Z planes and distort distances. This prompted us to develop a specialized software tool able to unwrap the signal of the roughly cylindrical cell layer of the aorta onto a plane (2D plane images will be referred to as 2D-maps thereafter). This software allows the treatment of information through time, taking into account heterogeneities and variations in the shape of the aorta along its axis using semi-automatic adjustment of the surface (see *Figure 3A* and Materials and methods for details). Using this software, applied mainly to eGFP-ZO1 TL sequences, we obtained a series of significant results.

Firstly, at 48 hpf, as exemplified in *Figure 3C–E* and *Figure 4*, the aortic circumference spans over three cells along most of its length.

Secondly, its cellular landscape is composed of two major populations of cells, displaying similar A-P length but a different aspect ratio; one spanning the aortic floor and distinctly narrower in the medio-lateral axis, and the other one spreading across the entire aortic wall, including occasionally the floor (*Figure 4B,D* and legend). In the TL sequence from which images presented in *Figure 3C–D* were extracted (see *Figure 3—video 1*), the cells of the former population (cells e1 and e2) underwent shrinkage along the A-P axis, and performed an EHT. These data support the idea that the laterally narrower cells spanning the aortic floor constitute the hemogenic endothelium.

Thirdly, hemogenic cells are capable of performing mitosis contemporarily to their EHT, as exemplified in *Figure 3C,D* in which cell e1 divided symmetrically (based on their morphology) along the A-P axis. After mitosis, the daughter cells continued their shrinkage along the A-P axis until their emergence. Mitosis during EHT took place in the aortic plane in 16% of our TL sequences, and in the sub-aortic space in 31% of them.

Fourthly, the junctions between EHT cells and their neighbors are characterized by increased density of eGFP-ZO1, particularly at the level of tricellular junctions and contacting surfaces that are oriented more or less aligned with the medio-lateral axis (*Figure 3C* (cells e1 and e2), *Figure 4* and *Figure 3—video 1*, *Figure 4—video 1*). During the EHT, these regions of high junctional density flanking the EHT cell converged while the cell apex constricted and merged (*Figure 4A* and *Figure 3—video 1*), confirming what we described in *Figure 2A*. At the end of the process, the junctional contacts newly established between adjoining endothelial cells merged into a single line of high eGFP-ZO1 density oriented perpendicular to blood flow (*Video 1*, t = 07:00, yellow arrowheads).

Fifthly, the bijunctional interfaces between endothelial cells that are not in contact with EHT cells and are oriented perpendicular to the A-P axis also often exhibit an apparent increase in eGFP-ZO1 density (*Figure 4A*). Hence, we are tempted to deduce that the increase in tight junction components at bi- and tricellular contacts is aimed at reinforcing membrane interfaces that are under high mechanical tension. This would be consistent with the proposed function of tight junctions in the cross-talk with VE-cadherin dependent control of cell-cell tension (*Tornavaca et al., 2015*).

Finally, the number of EHT surrounding cells does not vary much during the EHT, with an average of five contacting cells at the beginning, getting down to three at the end of the process (*Figure 4C*). In the case of EHT cells undergoing mitosis, the daughter cells were spread apart from each other by junctions being established between two adjoining endothelial cells (*Figure 3D*, blue arrow).

## Circumferential actin is organized anisotropically

Most cases of morphogenic movements of tissues during development involve changes in cell geometry that rely on the longitudinal, medial-apical and/or circumferential contraction of actomyosin

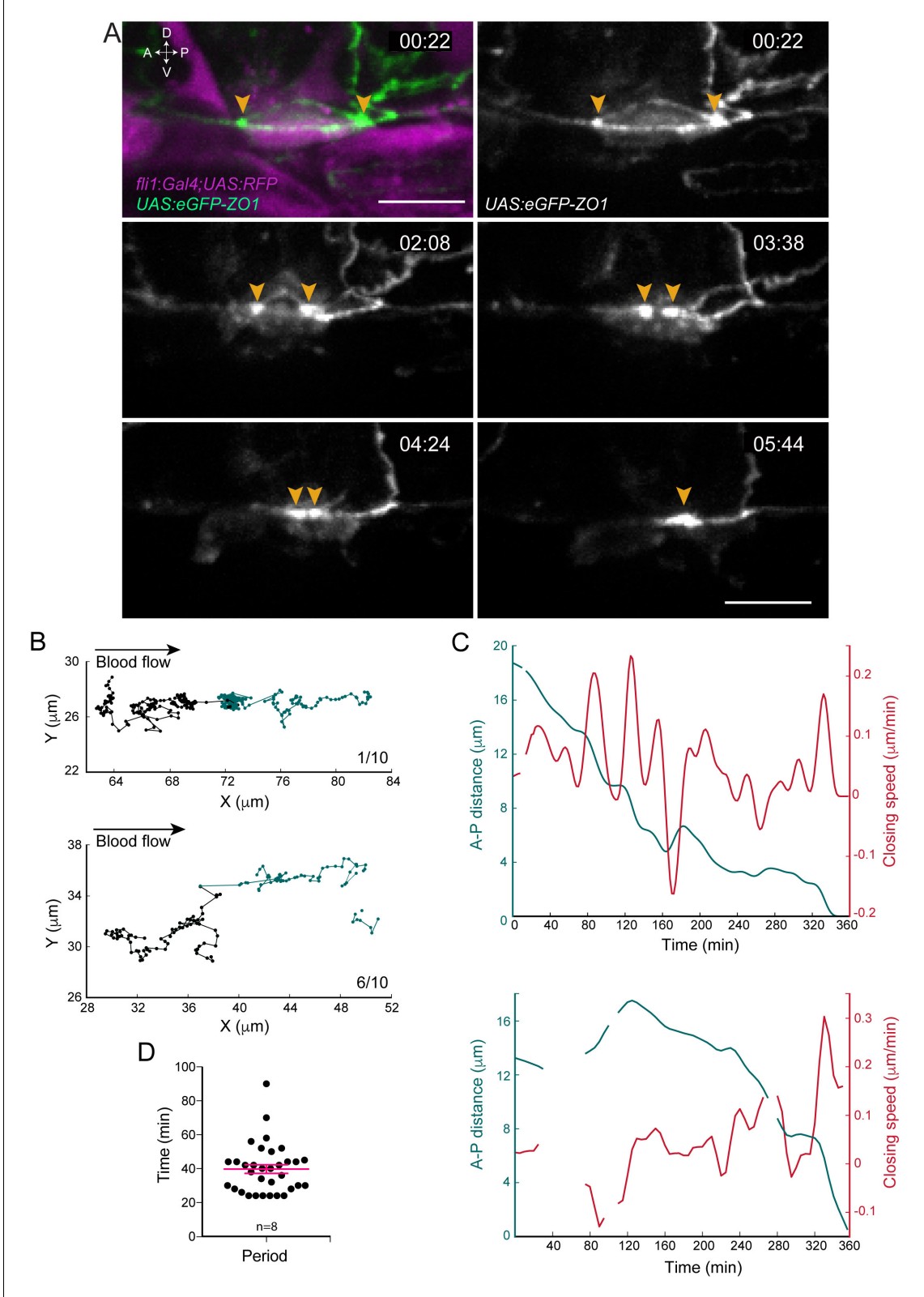

**Figure 2.** Apical constriction undergoes pulsatile activity. (**A**) Maximum projection of Z-planes extracted from a spinning-disk confocal TL sequence performed on a 48 hpf *Tg(fli1:Gal4; UAS:RFP; UAS:eGFP-ZO1)* embryo. Yellow arrowheads: regions enriched with eGFP-ZO1, at the two opposite poles of the EHT cell contacting endothelial neighbors and converging until coalescence (see *Figure 2—video 1*). Time is indicated in hr:min. Scale bars, 10 μm. (**B–C**) Spatio-temporal analysis of two cells (see also *Source code 1*). Top panels obtained from the cell visualized in (**A**) and bottom ones from a

*Figure 2 continued on next page*

*Figure 2 continued*

second cell. (B) Tracking of two spots delineating regions of eGFP-ZO1 densification, after maximum projection. Top and bottom panels are showing tracks of antero (black) and posterior (green) eGFP-ZO1 spots for a single cell each. X and Y correspond to the antero-posterior and dorso-ventral axis, respectively. Each point corresponds to the position in X,Y of the spot at a given time point. The lines link two successive time points. Note that when one spot is not detected, the line is interrupted. The anterior track is the longest track in 30% of cases, n = 10 (not shown); The posterior track is the longest track in 60% of cases, n = 10 (bottom panel); note that in 1/10 cells, the displacement of the antero and posterior spots were virtually equal (top panel). (C) Evolution as a function of time of distances (green) and closing speeds (red) of the two eGFP-ZO1 spots. Note that breaking of tracks and lines result from sporadic loss of signal. (D) Representation of the calculated periods of oscillatory closing speeds (39.7 min ±2.6 min). Error bars represent mean values ± SEM (n = 8 TL sequences).

DOI: https://doi.org/10.7554/eLife.37355.008

The following video and figure supplement are available for figure 2:

**Figure supplement 1.** Bbiomechanical parameters of the apical closure.

DOI: https://doi.org/10.7554/eLife.37355.009

**Figure 2—video 1.** Dynamics of the apical constriction followed by eGFP-ZO1.

DOI: https://doi.org/10.7554/eLife.37355.010

complexes composed of non-muscular myosin II (*Heisenberg and Bellaïche, 2013*; *Munjal and Lecuit, 2014*).

In order to investigate actomyosin function in the morphological changes taking place during the EHT, particularly in the contraction of the circumferential junctional belt, we first looked at the localization of filamentous actin (F-actin) in the vascular system, using Lifeact-eGFP as a reporter (Lifeact is a small non interfering peptide of 17 amino acids that allows visualization of actin dynamics and stains filamentous actin). We observed the recruitment of Lifeact-eGFP mainly at the cortico-apical side of the EHT cell, assembled as a belt at the interface with its endothelial neighbors with an apparent enrichment at its A and P poles (*Figure 5A*, upper two rows). The 2D-map algorithm revealed F-actin enrichment at contacting surfaces that are more or less perpendicular to the blood flow (*Figure 5A*, bottom panels). This localization is very similar to what we observed with eGFP-ZO1, suggesting that F-actin contributes to the organization of sub-junctional regions and perhaps the stabilization of junctional complexes that are, as proposed before, the most exposed to mechanical tension. In addition, the data also support the idea that the apical constriction of EHT cells requires the contractile activity of a circumferential actomyosin belt.

TL sequences performed with embryos expressing Lifeact-eGFP allowed us to follow the dynamics of cortico-apical circumferential actin (*Figure 5—video 1*, with its most informative images composing *Figure 5B*). This confirmed and extended our previous results, showing (i) an apical contractile actin belt that ultimately shaped into a contractile ring upon the narrowing down of the A-P luminal distance and (ii) the subsequent formation of an F-actin enriched foot that preceded the full release of the cell. We speculate that this latter pool of F-actin may result from the dissolution of the belt, and might serve as a propeller to trigger mechanical force required to complete the exit and/or the mechanical uncoupling of cell-cell junctions.

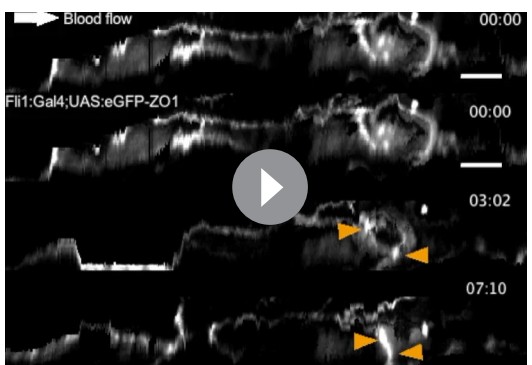

**Video 1.** Dynamic 2D-maping of the apical constriction followed by eGFP-ZO1. Spinning-disk TL confocal sequence performed on a 48 hpf *Tg(fli1:Gal4; UAS:RFP; UAS:eGFP-ZO1)* embryo, showing a cell undergoing the EHT and processed with the 2D-algorithm. Only the eGFP channel is shown. Note that this is a particular example of an EHT whose axis of emergence diverges by an angle of approximately 45°, from the A-P axis. Yellow arrowheads point at regions of high eGFP-ZO1 density (well visible at t = 03:02). Regions of high eGFP-ZO1 density converge (t = 03:32) to reach coalescence (t = 04:14). At the end of the process, the junctional contacts between adjoining endothelial cells merge into a single eGFP-ZO1-enriched line oriented perpendicular to blood flow. Time is indicated in hr: min. Scale bars, 10 μm.

DOI: https://doi.org/10.7554/eLife.37355.016

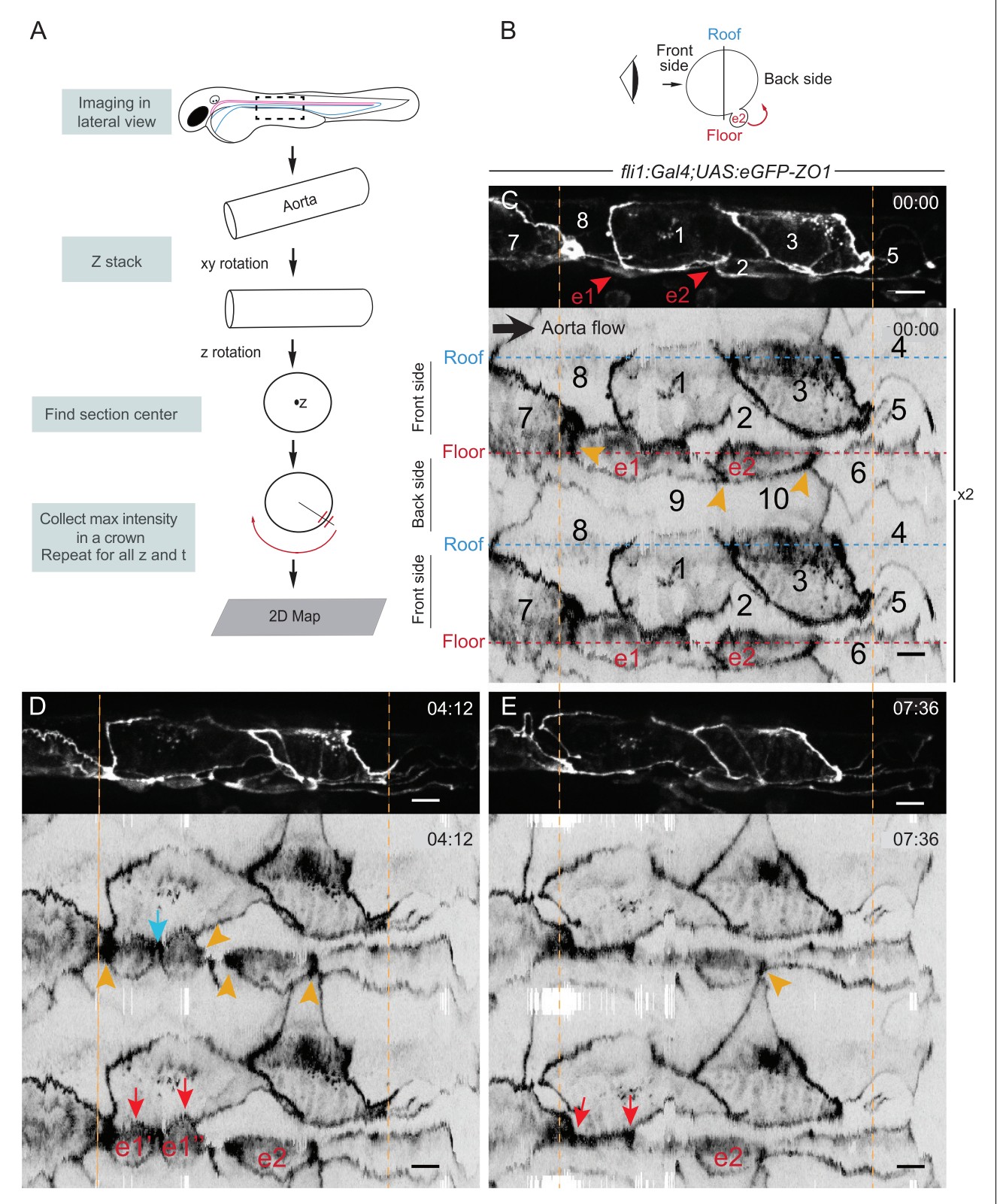

**Figure 3.** The EHT cellular landscape unravelled by 2D-projection. Schematic representations of: (**A**) Successive steps leading to the 2D projection (see text, Materials and methods and *Source code 2*), (**B**) The angle at which the Z-projections in the top panels (**C–E**) are visualized (eye) and indicating the back-sided exit of the EHT cell e2. (**C–E**) Top panels are maximum projections of Z-planes extracted from a spinning-disk confocal time-lapse performed on a 48 hpf *Tg(fli1:Gal4; UAS:RFP; UAS:eGFP-ZO1)* embryo. Bottom panels have been obtained after duplication (**x2**) of the corresponding
*Figure 3 continued on next page*

*Figure 3 continued*

2D-maps (see Materials and methods and *Figure 3—video 1*). Numbers indicate individual cells. e1, e2: EHT cells (in red, with e1 dividing into e1', e1''
by symmetric division, panel D red arrows) are located on the aorta floor. Cells 6, 9 and 10 are exclusively on the backside. All the others span mainly
the front side. Yellow arrowheads: Increased density of eGFP-ZO1 at contacting regions between EHT or endothelial cells that are oriented
perpendicular to blood flow (antero-posterior axis). Blue arrow: space between the two daughter cells. Note that in panel (**E**), the daughter cells have
emerged (red arrows). Time is indicated in hr:min. Scale bars, 10 µm.
DOI: https://doi.org/10.7554/eLife.37355.011
The following video is available for figure 3:
**Figure 3—video 1.** Dynamics of the EHT in the aortic landscape using 2D-maping-1.
DOI: https://doi.org/10.7554/eLife.37355.012

## Hemodynamic forces control the morphodynamics of the emergence

The anisotropic organization of F-actin and ZO1 in hemogenic and EHT cells, with their enrichment
at the antero-posterior poles more or less aligned with the blood flow axis, suggested that hemody-
namic forces may have an influence on the cytoskeleton architecture in these cells as well as on the
cytoskeletal/junctional interface. The involvement of blood flow in hematopoiesis was initially shown
in seminal work in the zebrafish (*North et al., 2009*) in which the abrogation of heart beating and
blood circulation impaired arterial identity and HSC formation (using the silent heart (sih) mutant of
cardiac troponin T [*Sehnert et al., 2002*]). Hence, to question the involvement of hemodynamic
forces on the cytoskeleton of hemogenic cells and, more broadly, the dynamics of HSC emergence,
we interfered with heart beating using a sih morpholino (injected in *Tg(kdrl:Ras-mCherry; kdrl:Life-
act-eGFP)* embryos to visualize F-actin and cellular membranes). We observed, in addition to major
alterations in the morphology of the dorsal aorta (*Figure 6A*), the recurrent absence of EHT cells
with cup-shaped morphology and emerging toward the sub-aortic space (14 embryos injected with
the sih morpholino were carefully analysed, all along the dorsal aortic wall). Rather, the aortic floor
was decorated with ovoid cells, with their luminal/apical membrane protruding into the lumen of the
aorta (*Figure 6B*). The lumen also contained cells labelled with Lifeact-eGFP and Ras-mCherry
(hence from vascular origin and supposedly having performed their extrusion from the surrounding
aortic wall since circulation is abrogated and cells in the aortic lumen do not migrate extensively in
comparison to cells in the sub-aortic space, see *Figure 6—video 1* for Z-sectioning of the aorta and
visualization of the content of the aortic lumen). TL sequences performed in the trunk region
revealed that the ovoid cells in the aortic floor do undergo mitosis and are able to emerge either
toward the sub-aortic space or, more unexpectedly, toward the aortic lumen (*Figure 6—video 2*,
right), which we never saw in the control condition (this intra-aortic emergence was observed in 3/3
movies obtained from three independent injection experiments). Interestingly, in one of our TL
sequences (1/3, the longest TL sequence that lasted 13.30 hr), two of the released cells ended up
bursting into pieces (see *Figure 6—video 2*, left), suggesting that cells released in the absence of
mechanical constraints exerted by blood flow are prone to cell death. This is consistent with previous
studies showing a reduction in the number of hematopoietic stem cells and progenitors upon sih
morpholino treatment (in particular in the caudal hematopoietic tissue, see Murayama et al., 2006;
*Murayama et al. (2006)*; *Bertrand et al., 2008*).

Altogether, these results support the idea that hemodynamic mechanical forces contribute to the
inward bending of the apical/luminal membrane of EHT cells (the cup-shaped cells) as well as the
direction of the emergence (toward the sub-aortic space). They also suggest that the biomechanics
of the emergence may impact downstream cellular functions and/or commitments that, if not ful-
filled, impede cell survival. Interestingly, when heart beating and blood flow were temporarily
arrested using tricaine methanesulfonate (at 1 mg/ml), the inward bending of the apical/luminal
membrane was not observed anymore and cells performing the emergence toward the sub-aortic
space maintained their path (10 embryos were observed, all along the dorsal aortic wall, between a
time window of 35–135 min after interruption of blood flow, data not shown). This result strongly
supports the idea that components of hemodynamic forces (among which the mechanical strain, the
force perpendicular to the flow axis acting on the vascular wall) are exerting a direct action on the
apical/luminal membrane bending.

Finally, we also investigated the effect of obliterating blood flow on F-actin and ZO1 organization.
We managed to obtain reasonably good images after 2D-maping with the *Tg(kdrl:Ras-*

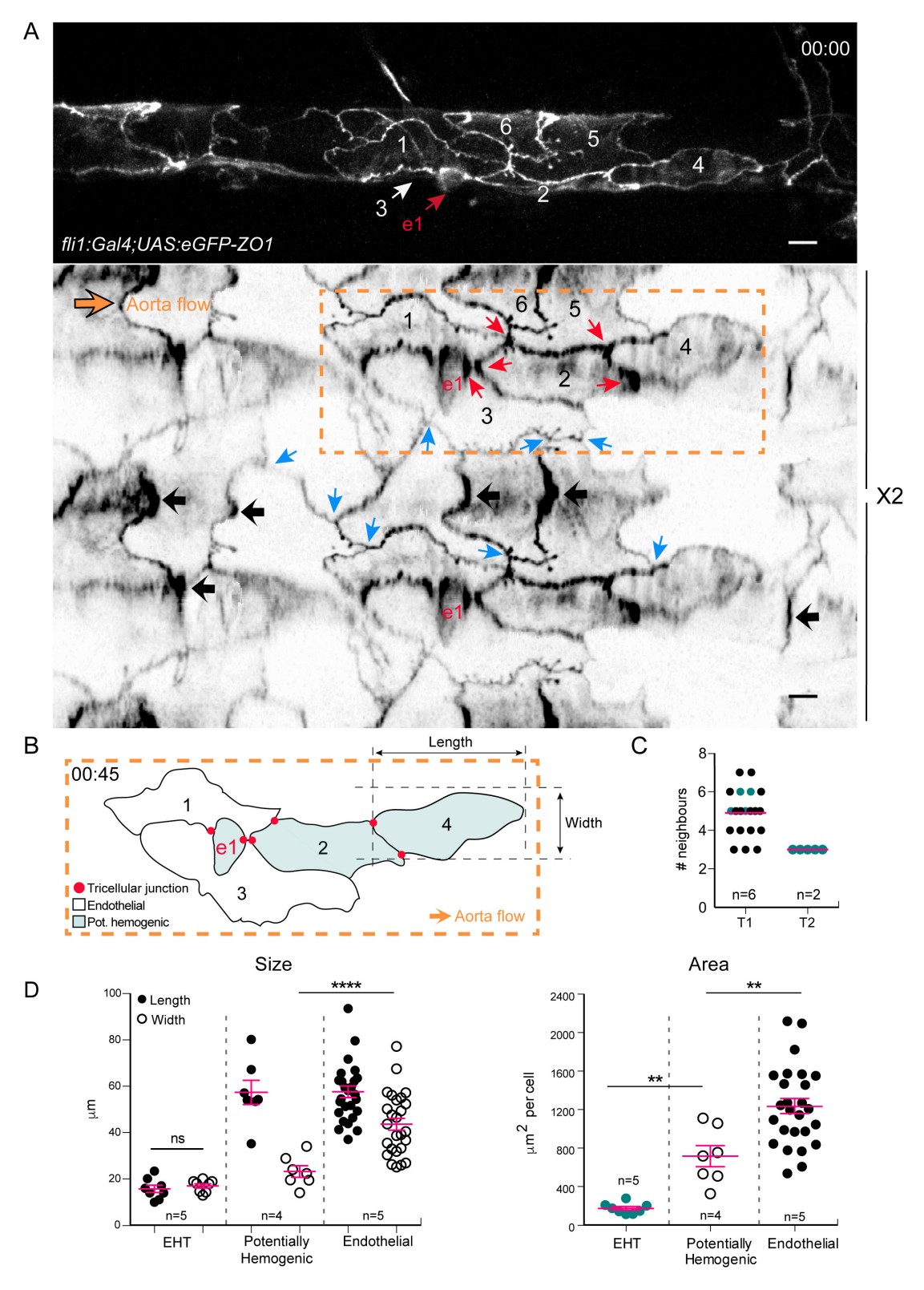

**Figure 4.** Intercellular contacts between hemogenic cells are reinforced. (**A**) Top panel: maximum projection of Z-planes extracted from a confocal TL sequence performed on a 48 hpf *Tg(fli1:Gal4; UAS:RFP; UAS:eGFP-ZO1)* embryo. e1; EHT cell, red arrow. Cell three localises on the back-side and partly surrounds the EHT cell (white arrow) (see also the 2D-map, bottom panel). Bottom panel: black arrows point at bicellular interfaces between endothelial cells, enriched with eGFP-ZO1 and perpendicular to the aorta flow. Blue arrows: tricellular junctions between endothelial cells. In the
*Figure 4 continued on next page*

*Figure 4 continued*

hemogenic region (yellow dotted rectangle), the red arrows point at tricellular junctions between potentially hemogenic cells and the EHT cell (e1, red) that are enriched with eGFP-ZO1 (see *Figure 4—video 1*). Scale bars, 10 µm. (**B**) Schematic representation of the hemogenic region in (**A**) emphasizing on the localization of tricellular junctions mainly between the EHT cell and a neighboring potentially hemogenic cell (e1, 2) and between two potentially hemogenic cells (2, 4). See Materials and methods for the definition of cell categories. (**C**) Number of cells surrounding EHT cells calculated from 2D-maps (see Materials and methods) at an early time point of the process (beginning of the TL sequence; T1, n = 6 maps), and at a late time point contemporary to emergence (T2, n = 2 maps). The green points represent values calculated from the 2 TL sequences for which emergence could be captured. Red bars, median. (**D**) Morphometric analysis of EHT cells at advanced cup-shaped stage, potentially hemogenic and endothelial cells calculated from 2D-maps (see also *Figure 4—source data 1*). Left: Size. The EHT cells advanced in the process exhibit nearly equal length (15.7 ± 1.6 µm) and width (17.0 ± 0.9 µm), potentially hemogenic cells that are elongated in the antero-posterior axis (length and width of (57.4 ± 5.2 µm) and (23.2 ± 2.5 µm)) and endothelial ones that have a larger width (length and width of (57.7 ± 2.5 µm) and (43.6 ± 2.6 µm)). Right: Area. Error bars represent mean values ± SEM. Unpaired t test (**) p<0.01; (****) p<0.0001 (n = number of maps).

DOI: https://doi.org/10.7554/eLife.37355.013

The following video and source data are available for figure 4:

**Source data 1.** *Figure 4D* Numerical data (width, length and area) and corresponding 2D-Maps with the contours of EHT cells (red), hemogenic cells (blue) and endothelial cells (green).
DOI: https://doi.org/10.7554/eLife.37355.014
**Figure 4—video 1.** Dynamics of the EHT in the aortic landscape using 2D-maping-2.
DOI: https://doi.org/10.7554/eLife.37355.015

*mCherry; kdrl:Lifeact-eGFP)* fish line, although with difficulties because of the strong distortion of the aortic shape in the sih embryos. Images clearly revealed a perturbation of F-actin organization in EHT cells, with little reinforcement of that staining at the antero-posterior poles of the cells in comparison to the control (*Figure 6B*, bottom panels). Unfortunately, in the case of eGFP-ZO1, fluorescence signals in the sub-cortical regions of hemogenic cells were too low and aorta too distorted to perform 2D-maping. However, 3D-projections and 3D-analysis with the Imaris software also revealed changes in eGFP-ZO1 localization in cells of the hemogenic endothelium, with altered recruitment at the sub-plasmalemmal level and apparent increase of the eGFP-ZO1 cytoplasmic pool (*Figure 6C* and see *Figure 6—video 3* for 3D-Imaris rendering). Altogether, these results support the idea that hemodynamic forces exert an action on the cytoskeletal architecture of hemogenic cells and impact on junctional organization.

## EHT sequential steps and key features

Time-lapse imaging of Lifeact-eGFP in embryos in which the EHT cells were best aligned with the A-P axis also suggested an active role of the adjoining endothelial cells in the process. The A and P neighbors moved toward each other over the actin-rich poles and appeared to extend actin-rich filopodia to establish contacts. Subsequently, filopodia anastomosis should contribute to the sealing of the aorta floor (*Figure 7—video 1*, with the main sequential steps composing *Figure 7*, left panels).

The live-imaging data presented so far led us to define the main, sequential steps of the EHT (see the cartoons illustrating steps 1–5, *Figure 7*, middle and right panels, and details in the legend). We propose that the key features of the process are: (i) the initiation of the bending of an hemogenic cell toward the sub-aortic space (with its antero-posterior axis more or less aligned in the direction of blood flow and the inward bending of its luminal/apical membrane depending on hemodynamic forces); (ii) the anisotropic organization of circumferential F-actin and of junctional contacts influenced by the blood flow (with enrichment at trijunctions and membrane interfaces perpendicular to blood flow/aorta A-P axis); (iii) the anisotropic, stepwise and oscillatory constriction of the cell apex, accompanied by concomitant contraction of a circumferential F-actin belt (essential molecular components of this contractile activity are myosin and associated light chains, see below); (iv) the apparent deployment of pushing forces by adjoining endothelial cells possibly contributing to cellular bending and apical constriction; (v) the extension of filopodia by adjoining endothelial cells; (vi) the anastomosis of filopodia to seal the aortic floor; (vii) the release of the EHT cell after remodelling of the luminal membrane (see Discussion) and down-regulation of remaining junctional complexes.

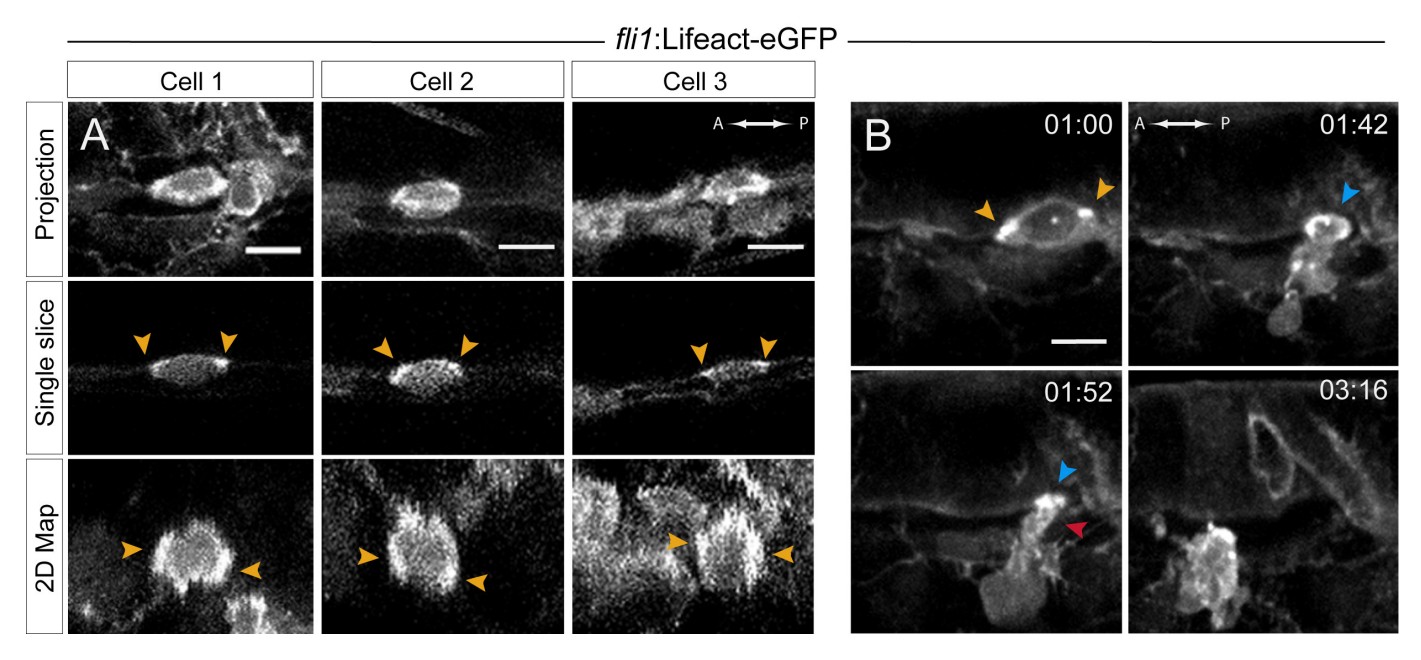

**Figure 5.** Actin localization and dynamics. (A–B) Lifeact-eGFP expressing 48 hpf embryos were imaged by spinning-disk confocal microscopy to obtain maximum projections of Z-planes (A, top and B), single Z planes (A, middle), or 2D-map projections (A, bottom; note that cells #2 and #3 are from the same aorta; see also *Figure 5—source data 1*). Yellow arrowheads: spots enriched with F-actin and aligned with the antero-posterior axis. (B) Images extracted from *Figure 5—video 1*. F-actin is enriched at the level of the contracting circumferential belt (top left) that converts into a ring (top right, blue arrowhead), and subsequently in the foot (red arrowhead). Time is indicated in hr:min. Scale bars, 10 μm.

DOI: https://doi.org/10.7554/eLife.37355.017

The following video and source data are available for figure 5:

**Source data 1.** *Figure 5A*: Z-projections (on top) and full 2D-Maps (bottom) from which the images presented in the bottom panels of *Figure 5A* were extracted.

DOI: https://doi.org/10.7554/eLife.37355.018

**Figure 5—video 1.** Actin dynamics during the EHT.

DOI: https://doi.org/10.7554/eLife.37355.019

## Myosin regulatory light chains 9a and 9b are required for definitive hematopoiesis

We then addressed the mechanism of contractility of the circumferential actin belt by investigating the role of myosin and, more specifically, the regulation of its activity by myosin regulatory light chains (RLCs). We focused on the regulatory light chain 9 (Myl9) because the *myl9* gene was shown to be a target of Runx1 in a genome-wide analysis of the transcriptional profile and of Runx1 binding in a model of the mouse hemogenic endothelium in cell culture (*Lie-A-Ling et al., 2014*). Owing to gene duplication in teleosts, the *myl9* gene was duplicated in the zebrafish, leading to the expression of two paralogs bearing more than 93% identity and referred to as Myl9a and Myl9b (for sequence comparison, see *Figure 8—figure supplement 1A*). By WISH, we confirmed that the *myl9a* mRNA is expressed in the dorsal aorta of 48 hpf embryos (*Figure 8A*) and we found that so is the case for *myl9b*. More precisely, we detected the *myl9b* mRNA in the brain, somites, and dorsal aorta (owing to strong signals in the ventral part of the somites, transverse sectioning was necessary to visualize unambiguously the localization of *myl9b* mRNA in aortic cells, *Figure 8A*, right panel).

We next investigated the incidence of decreasing the expression level of Myl9a or Myl9b on definitive hematopoiesis. We designed two pairs of splice-blocking morpholinos targeting each mRNA (morpholino efficiencies were validated by qPCR, see *Figure 8—figure supplement 1B,C*), and injected them in *Tg(CD41:eGFP)* embryos. The overall development and blood circulation of injected embryos appeared normal. GFP-positive cells were then scored at 48 hpf, in the sub-aortic space (AGM) (to be most proximal to the site of HSPCs emergence), and in the caudal hematopoietic tissue

(CHT), the first niche where aorta-derived HSPCs home. In both knock-downs, we observed a significant decrease in HSPC number in the AGM and the CHT (*Figure 8B–E*). This phenotype was phenocopied by two additional morpholinos targeting myl9b and myl9a mRNAs respectively and was rescued for the spe3i3 myl9b morpholino (*Figure 8—figure supplement 2*). These data support the involvement of both Myl9a and Myl9b in definitive hematopoiesis, and possibly in the EHT process.

Finally, in order to address the intracellular localization and dynamics of the two Myl9 isoforms during the EHT, we generated *Tg(kdrl:Myl9a-eGFP)* and *Tg(kdrl:Myl9b-eGFP)* fish lines. Since the fluorescence intensities of Myl9a and Myl9b fused to eGFP were much lower than in the case of Lifeact-eGFP, we were unable to treat our movies with the 2D algorithm. However, as seen in *Video 2*, regular TL sequences show convincingly that Myl9a-eGFP and Myl9b-eGFP assemble as cortico-apical belts with enrichment at spots between EHT cells and neighbors that move, upon apical constriction, along the A-P axis, and coalesce. This is reminiscent of what we observed with Lifeact-eGFP and eGFP-ZO1 (see *Video 3* for a comparative 'en face' visualization of the dynamics of actin and myosin belts/rings). We also noticed that Myl9a and b were recruited at the contracting cytokinesis ring in EHT cells that underwent mitosis (see *Video 2* and its legend).

## The amino-terminal phosphorylation site on Myl9b is essential for definitive hematopoiesis

Myl9b and Myl9a are very similar (93% identity calculated by ClustalO), and the most significant difference resides in their amino-terminus, where the two serines following the starting methionine (which are strictly conserved in mammals and in most zebrafish RLCs, see *Figure 8—figure supplement 1A*) are present in Myl9b, but not in Myl9a. These two residues are the substrates of protein kinase C (PKC) activity (*Ikebe et al., 1987*). It was shown that their phosphorylation impairs the phosphorylation of the downstream Rho-kinase/MLCK/Citron-kinase target site (*Figure 8—figure supplement 1A*) that is essential for the stimulatory activity of RLCs on the ATPase activity of the myosin heavy chain (*Turbedsky et al., 1997*). Hence, phosphorylation of this amino-terminal site inhibits myosin activity and actomyosin contraction.

We mutagenized the Myl9b cDNA to substitute the two amino-terminal serine residues by alanines, and expressed the wt or mutant form (Myl9bA2A3) fused to eGFP in zebrafish embryos by

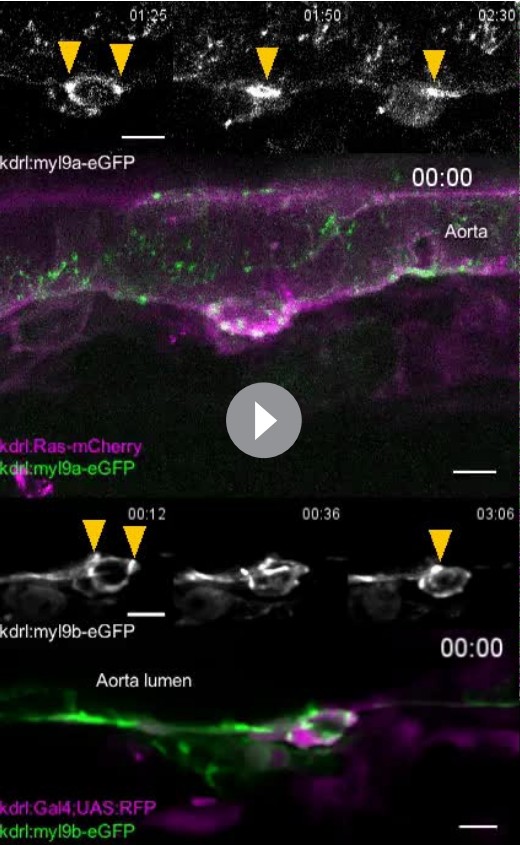

**Video 2.** Dynamics of Myl9a and Myl9b during the EHT. Top panels: Maximum projection of Z-planes of a laser scanning confocal TL sequence performed on a 48 hpf *Tg(kdrl:Ras-mCherry; kdrl:Myl9a-eGFP)* embryo focusing on the localization of Myl9a-eGFP at the circumferential belt and its accumulation at the two opposite poles aligned with the A-P axis (t = 01:25). Note the progression of belt closure (t = 01:50) and the coalescence of the two points at the end of the EHT (t = 02:30). Note the independent migrating cell coming into contact with the EHT cell (t = 01:40). Note also that the cell divides after the release and that Myl9a-eGFP is recruited to the cytokinesis ring (white arrow, t = 02:55). Time is indicated in hr:min. Scale bars, 10 μm. Bottom panels: Maximum projection of Z-planes of a spinning-disk confocal TL sequence performed on a 48 hpf *Tg(kdrl:Gal4; UAS:RFP; kdrl:Myl9b-eGFP)* embryo focusing on accumulation of Myl9b-eGFP at the cell borders connecting with neighboring cells (t = 00:12) and at a luminal circumferential belt that coalesces at the end of the EHT (t = 00:36 and 03:06 respectively, yellow arrowheads). Note that the cell divides after the release and that Myl9b-eGFP is recruited to the cytokinesis ring (white arrow at t = 08:56). Time is indicated in hr:min. Scale bars, 10 μm.

DOI: https://doi.org/10.7554/eLife.37355.029

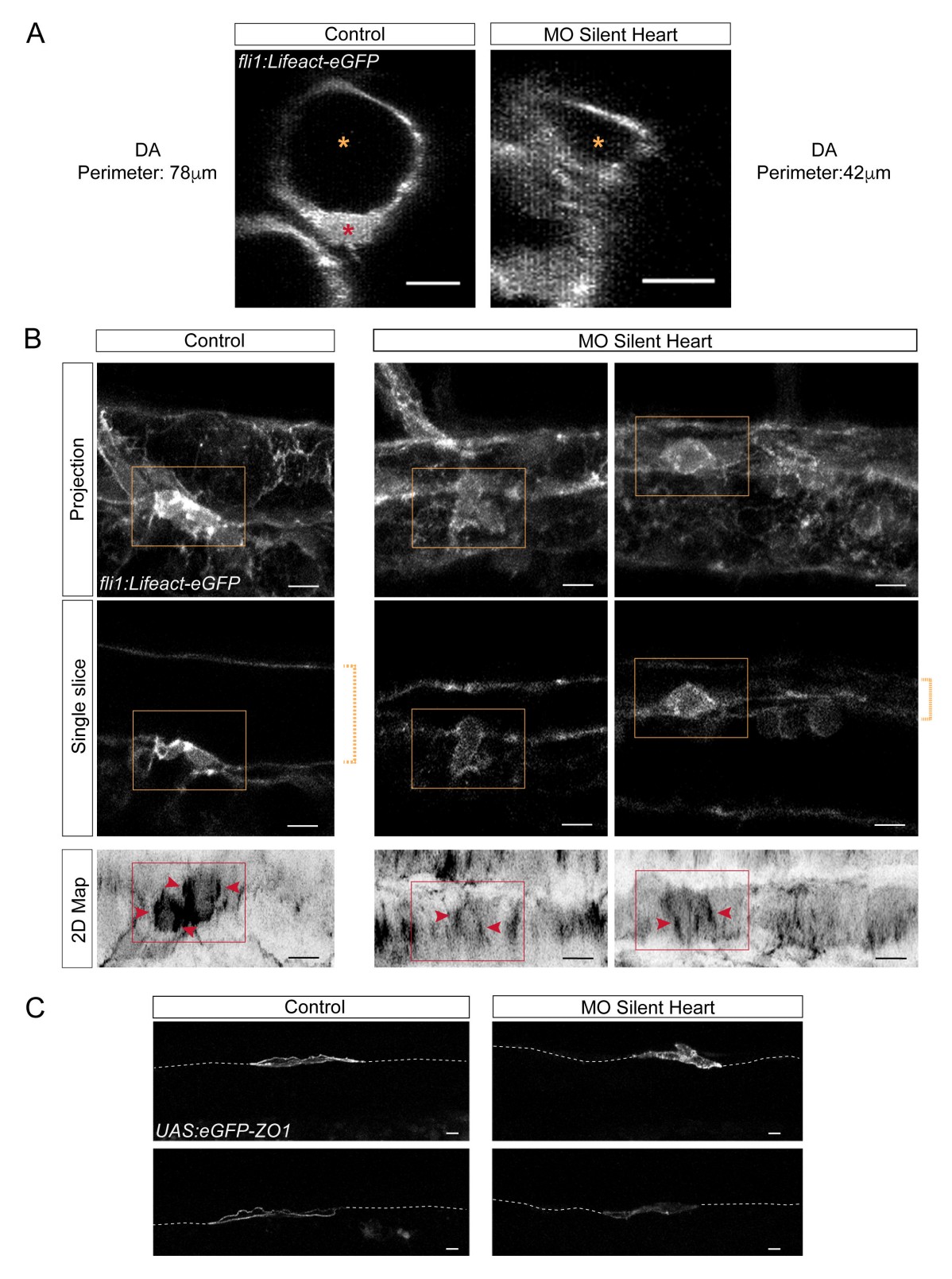

**Figure 6.** Blood flow influences the morphology as well as the cytoskeletal and junctional organization of hemogenic and emerging cells. Lifeact-eGFP (**A,B**) or eGFP-ZO1 (**C**) expressing 48 hpf embryos (**A,B**) were imaged by confocal microscopy after injection of sih or control morpholinos. (**A**) Transversal views of control and sih morphants show the flattening of the aorta and the decrease in perimeter when blood flow is abrogated. Yellow stars indicate the aortic lumen and the red star an EHT cell emerging from the ventral floor. (**B**) Maximum projections of Z-planes (top panels), single Z
*Figure 6 continued on next page*

*Figure 6 continued*

planes (middle panels and see *Figure 6—video 1*), or 2D-map projections (lower panels). Dashed yellow brackets highlight the thickness of the aorta. Yellow rectangles surround EHT undergoing cells. The same cells visualized after 2D projections are surrounded by red rectangles. Red arrowheads point at actin densification, in EHT cells, aligned with the antero-posterior axis of the dorsal aorta. (C) Maximum projections showing eGFP-ZO1 localization in two hemogenic control and sih morphants cells lying on the aortic floor. Note the mosaicism owing to transient expression of the UAS: eGFP-ZO1 transgene (for a 3D-rendering view and visualizing the aortic contours, see *Figure 6—video 3*). Dashed lines indicate the position of the aortic floor. Scale bars, 10 μm.

DOI: https://doi.org/10.7554/eLife.37355.020
The following videos are available for figure 6:
**Figure 6—video 1.** Aortic phenotype in sih morphants.
DOI: https://doi.org/10.7554/eLife.37355.021
**Figure 6—video 2.** Morphodynamics of the emergence in sih morphants.
DOI: https://doi.org/10.7554/eLife.37355.022
**Figure 6—video 3.** Alteration of ZO1 localization in sih morphants.
DOI: https://doi.org/10.7554/eLife.37355.023

transient transgenesis. Expression of the mutant form using the kdrl promoter caused mild morphological alterations of the dorsal vascular system (with maintenance of blood flow, no obvious change in the shape of the dorsal aorta and a few distortions at the base of intersegmental vessels, data not shown). To restrict the expression of Myl9-bA2A3 or the wt form to aorta-derived HSPCs, we used the Runx'1 + 23' enhancer (*Tamplin et al., 2015*), which we confirmed to drive transient expression specifically in these cells (*Figure 9—figure supplement 1*). Using this promoter, we transiently expressed Myl9b-eGFP and Myl9bA2A3-eGFP in *Tg(Kdrl:Ras-mCherry)* embryos to visualize Myl9b-expressing HSPCs together with the vascular system (*Figure 9*). At 40 and 50 hpf, and in comparison with the wt form, we found a much lower number of HSPCs expressing the mutant form, both in the sub-aortic space and in the CHT (*Figure 9A,B*). These results suggest that preventing N-terminal phosphorylation impairs definitive hematopoiesis and may interfere with the emergence from the aortic floor. Finally, we performed live imaging in this context, and made a series of observations. Firstly, while we occasionally found hemogenic cells in the aortic floor that expressed the wt Myl9b-eGFP and underwent EHT, we were unable to find such cells upon expressing Myl9-bA2A3-eGFP, suggesting that the latter cells emerged at a significantly lesser frequency. Secondly, cellular fragments were often seen in the sub-aortic area, usually indicative of cell bursting. This cell fragmentation may have occurred during the EHT as described upon *runx1* knock-down (*Kissa and Herbomel, 2010*), or in EHT-derived HSPCs as upon scl-α knock-down (*Zhen et al., 2013*). An example of cell fragmentation probably taking place short after the emergence is shown in *Figure 9C* and *Figure 9—video 2*. Thirdly, EHT-derived HSPCs expressing

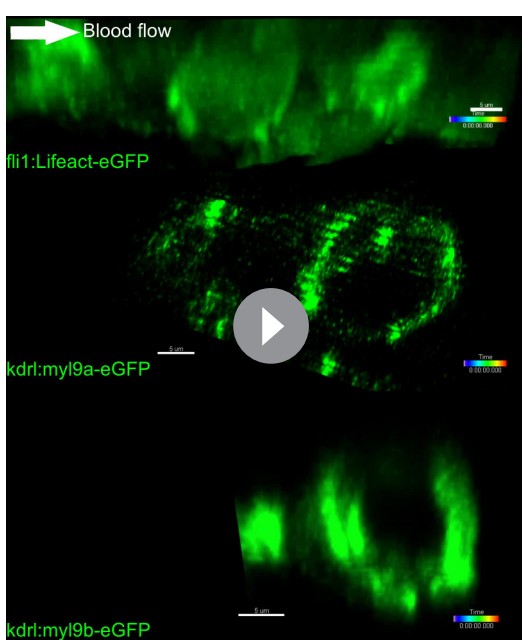

**Video 3.** Dynamics of the contractile and anisotropic actin and myosin rings. Spinning-disk confocal TL sequence showing 3D rendering 'en face' views of EHT cells from 48 hpf *Tg(fli1:Lifeact-eGFP)* (top panel), *Tg(kdrl:Myl9a-eGFP)* (middle panel) and *Tg(kdrl:Myl9b-eGFP)* (bottom panel) embryos. White arrows point at Lifeact-eGFP (t = 01:46), Myl9a-eGFP (t = 01:30) and Myl9b-eGFP (t = 02:22) accumulations at the two opposite poles of the EHT cells that are more or less perpendicular to blood flow and aligned with the A-P axis (white arrowheads are positioned parallel to that axis). Note the conserved anisotropy of the belt through time and the dynamics of its closure. Time is indicated in hr:min. Scale bars, 5 μm (top and bottom panels) and 10 μm (middle panel).
DOI: https://doi.org/10.7554/eLife.37355.030

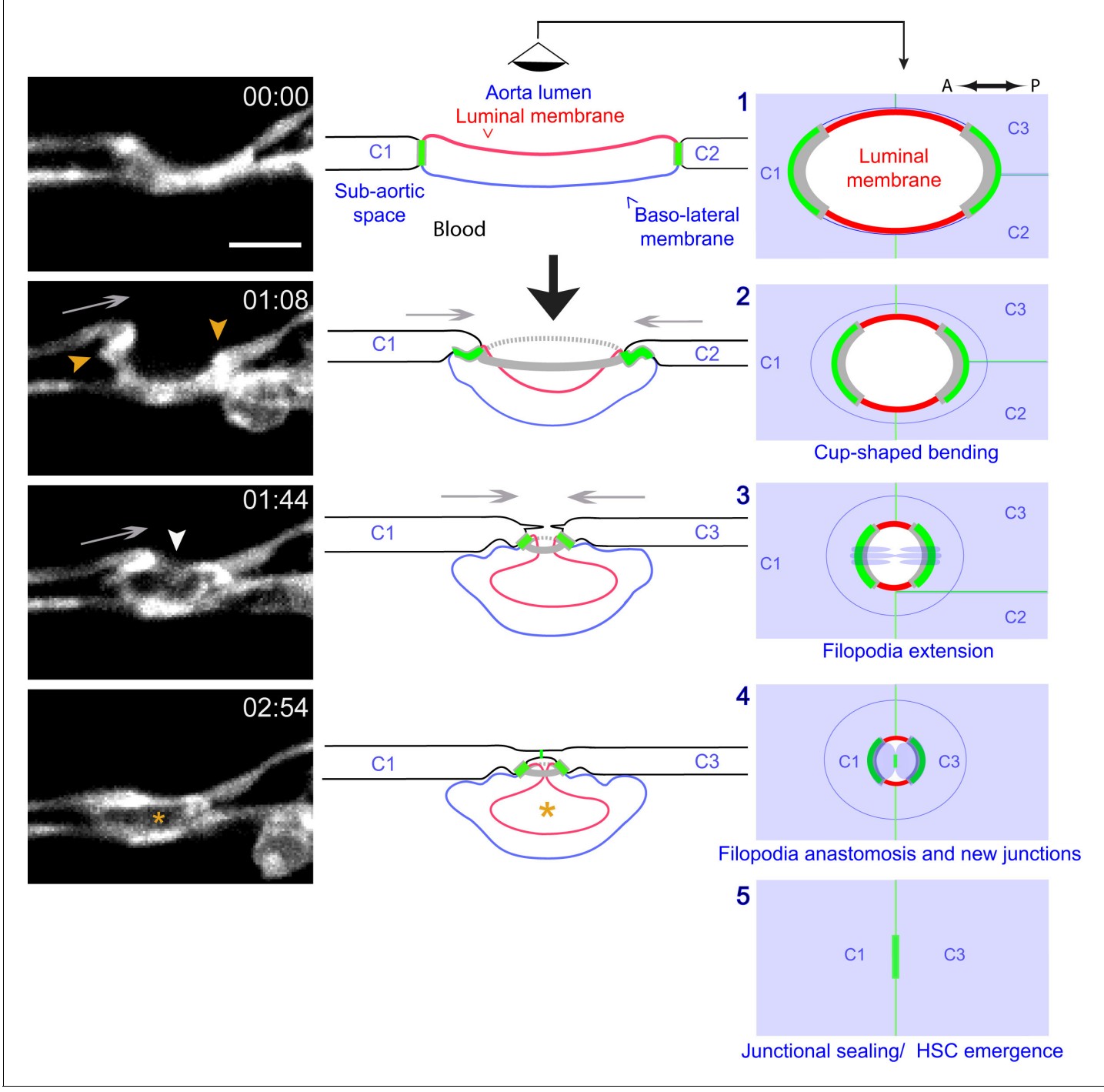

**Figure 7.** Modeling of the EHT process. Left panels: Lifeact-eGFP expressing 48 hpf embryo imaged by spinning-disk confocal microscopy to obtain single Z planes. Images were extracted from *Figure 7—video 1*. Grey arrows: evidence for the deployment of pushing forces by adjoining endothelial cells. Yellow arrowheads: Lifeact-eGFP densifications. White arrowhead: extension and convergence of filopodia hypothetically emanating from an endothelial cell. Asterisk: intracellular cavity resulting from invagination of the luminal membrane. Middle and right panels (lateral views and 'en face' views (with the eye positioned in the aorta lumen)): schematic drawings representing five sequential steps of the EHT deduced from the localization of eGFP-ZO1 (green lines: junctional interfaces), Ras-mCherry (red and blue: luminal and basal membrane, respectively), Lifeact-eGFP (grey, F-actin circumferential belt. Note that the junctional circumferential belt is not drawn for simplicity). Grey arrows: putative pushing forces deployed by adjoining endothelial cells. Black arrow: hemodynamic force (perpendicular to the aortic plane). Note that in the 'en face' views, the luminal membrane should appear as a red surface (depicted by red lines only), and that the circumferential junctional and F-actin belts are only depicted at their sites of

*Figure 7 continued on next page*

Figure 7 continued

increased density, that is membrane interfaces at the A-P poles of the EHT cell, bound by tricellular junctions oriented perpendicular to blood flow, as revealed by 2D-maps. See the main text for additional comments. C1-C4: adjoining endothelial cells. Time is indicated in hr:min. Scale bar, 10 µm.
DOI: https://doi.org/10.7554/eLife.37355.024
The following video is available for figure 7:
**Figure 7—video 1.** Evidence for forces deployed during the EHT.
DOI: https://doi.org/10.7554/eLife.37355.025

Myl9bA2A3-eGFP exhibited abnormal shapes and migration, characterized by anarchic membrane extensions in many directions at once (*Figure 9D* and *Figure 9—video 1*). Interestingly, they were still able to undergo mitosis, with Myl9bA2A3 relocating to the cytokinesis ring as wt Myl9b would do (data not shown).

## Discussion

This study led us to envision the adaptation of the EHT to constraints imposed by the aortic environment, in particular the mechanical forces exerted by the blood flow. It strengthens the idea that the process is indeed very unique when compared to other single cell extrusion or delamination processes described so far. The high spatio-temporal resolution of our TL sequences and their qualitative and quantitative analysis, in particular using an appropriate 2D-algorithm, highlight the specific features of the EHT, schematized in *Figure 7*. Here, in addition to underlining the peculiarities of the EHT, we propose to discuss the results that are most relevant to a comparative analysis of the process with other cellular extrusion mechanisms.

### The bending of the apical/luminal membrane and the orientation of the emergence

The inward bending of the luminal membrane of EHT cells is one of the hallmarks of the EHT that, to our knowledge, has never been described so far for any other type of cell extrusion. In the context of the EHT, the direction of extrusion is opposite depending on the species, i.e towards the sub-aortic space in the zebrafish and the aortic lumen in the mouse (*Klaus and Robin, 2017*). Although the two emergences proceed in opposite directions, it appears that the EHT, in these two species, may have something in common. Indeed, electron microscopic images of cells of the embryonic mouse aorta floor showed the presence of large intracellular vacuole-like structures (*Marshall and Thrasher, 2001*) reminiscent of what is observed in the zebrafish embryo, raising the idea that the inward bending of the apical/luminal membrane may be followed by its intracellular release after the emergence.

If the reason for the discrepancy regarding the direction of the emergence between the mouse and the zebrafish is currently unknown, this study contributes shedding light on this issue. When abrogating the blood flow by obliterating heart beating from the very beginning, we show that cells from the aortic floor can emerge in the aortic lumen (presumably hemogenic cells endowed with the potential to become hematopoietic). Unless, in the zebrafish, hemogenic cells that have never been exposed to hemodynamic forces do not express the relevant cytosolic machineries essential for performing a sub-aortic emergence, this raises the possibility that the variation between the two species is a direct consequence of differences in the strength of components of hemodynamic forces, such as, for example, the mechanical strain (the force perpendicular to the direction of the flow and that relates to the rhythm of heart beating, *Hahn and Schwartz, 2009*). However, the differences in heart beating frequencies between the two species are not so far from each other at that developmental stage (approximately 2–3 beats/s, [*Xing et al., 2018*; *Hahurij et al., 2014*]). Hence, it is not so clear if the sub-aortic orientation of the emergence of HSCs in the zebrafish may be a direct consequence of the mechanical strain; what might count as well regarding the difference between the two species are the physical properties of the respective aortic walls that should impact the way they respond to the force because of their diameters, much wider in the mouse than in the zebrafish (approximately 300 µm in the mouse and 25 µm in the zebrafish embryos at the time window of the EHT).

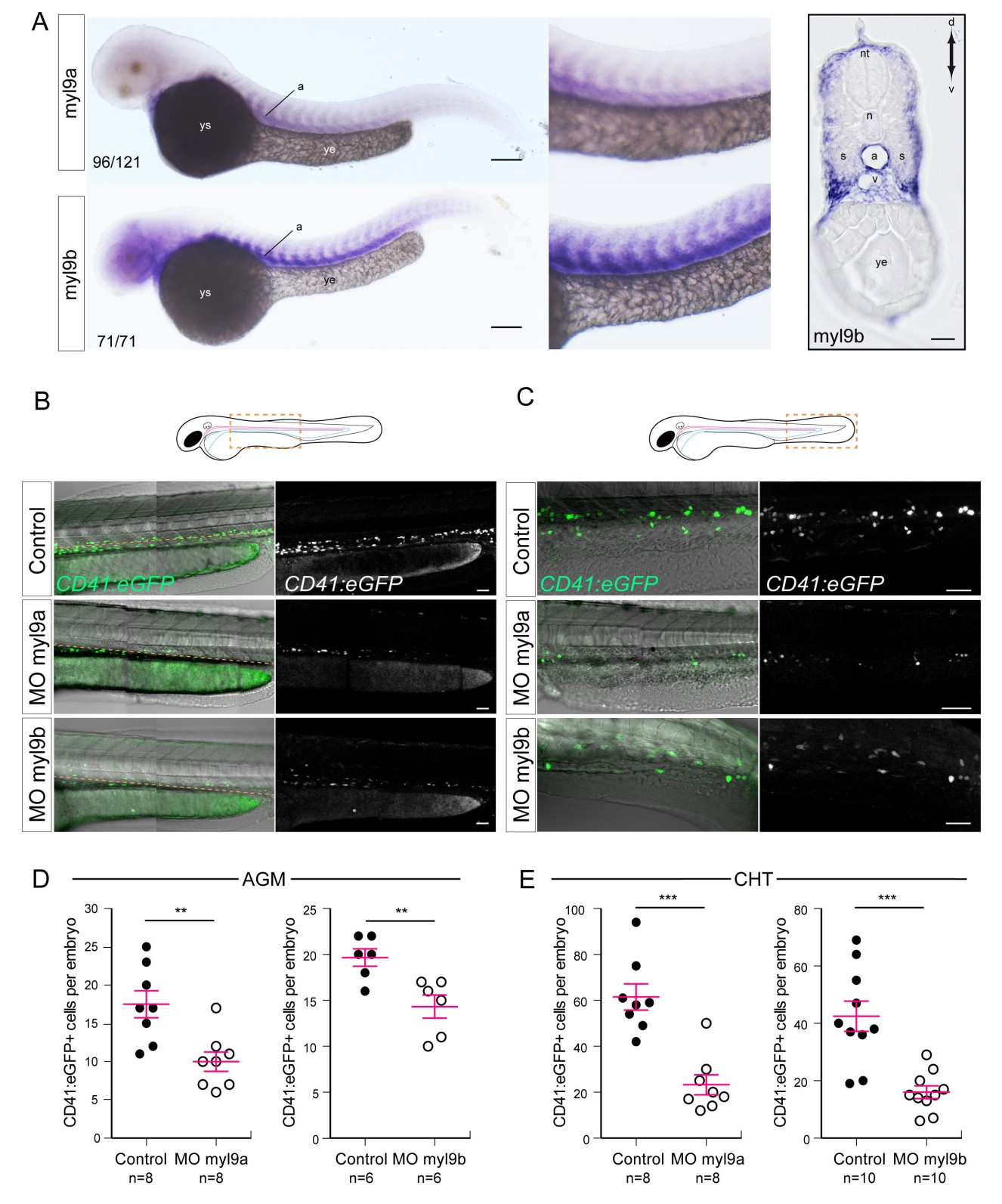

**Figure 8.** Myosin regulatory light chains 9a and 9b are required for definitive hematopoiesis. (**A**) In situ hybridizations of 48 hpf embryos. Left panels, whole mount; right panel, transversal section. a, aorta; n, notochord; nt, neural tube; s, somites; v, vein; ye, yolk extension; ys, yolk sac. Results were obtained from four independent experiments. Note that for the 25/125 embryos in the top panel, the myl9a signal in the aorta was weaker and masked by the somitic staining. (**B,C**) Maximum projection of Z-planes obtained from 48 hpf *Tg(CD41:eGFP)* embryos, focusing either on the AGM region (**B**), or

*Figure 8 continued on next page*

*Figure 8 continued*
on the CHT (C). Scales bars, 50 µm (B); 100 µm (C). Only the CD41:eGFP⁺ cells above the yellow dashed line were counted (C), as those beneath are
not necessarily hematopoietic cells (*Kissa et al., 2008*). (D,E) Effect of myl9a and myl9b morpholinos (e2i2 and e3i3, respectively, see Materials and
methods for details and *Figure 8—figure supplement 1*) based knockdown on the number of CD41:eGFP⁺ cells in the AGM (D) and the CHT (E).
Similar results, in the AGM and in the CHT, were obtained at lower quantities of morpholinos (see *Figure 8—figure supplement 2*). n = indicate the
numbers of embryos. (B, E, F) Error bars represent mean values ± SEM. Unpaired t test (**) p<0.01; (***) p<0.001.
DOI: https://doi.org/10.7554/eLife.37355.026
The following figure supplements are available for figure 8:

**Figure supplement 1.** Quantitative analysis of the effects of myl9a and b splicing morpholinos (see also text).
DOI: https://doi.org/10.7554/eLife.37355.027
**Figure supplement 2.** Myl9a and myl9b morpholinos phenocopy of hematopoietic phenotype and myl9b morpholino rescue.
DOI: https://doi.org/10.7554/eLife.37355.028

## Mechanical forces exerted by blood flow control the EHT biomechanics

The period of the aortic development during which the EHT takes place is the one at which aortic cells are highly exposed to mechanical tensions. Indeed, it has been shown by using a VE-cadherin mechanical sensor, that young aortic cells (at 48 hpf), are more exposed to tension than 2 days later, when the aorta has matured and the aortic cells align with the direction of the flow (*Lagendijk et al., 2017*). Hence, elongated hemogenic and EHT cells, on the aortic floor, may also orientate and contract along that same axis (as we show in our work with the 2D-maping results), to minimize environmental constraints and membrane tension. In this context, the membrane interfaces that remain the most exposed to mechanical tensions are most probably the ones that are enriched in ZO1 and actin and are oriented perpendicular to blood flow (consistently with ZO1 enrichment at the interface between endothelial cells that spread over a large surface of the aorta and whose orientation is also perpendicular to blood flow). According to our results obtained upon abrogation of heart beating and showing alteration of actin anisotropy (i.e the reduction in the recruitment of sub-cortical actin at the two poles of the cells more or less aligned with the blood flow axis), tensions exerted by the hemodynamics control sub-plasmalemmal cytoskeletal architecture in EHT cells and, consequently, the actin/junctional interface (as suggested by the delocalization of ZO1 that we observe in these conditions). Altogether, these results support the conclusion that the EHT biomechanics are specifically adapted to - and influenced by - the mechanical tensions exerted by blood flow. It should be noted that this control by the flow might be direct (ex: via cell surface mechanosensors), and/or indirect (ex: via the induction of Runx1, the transcription factor essential for hematopoiesis [*Adamo et al., 2009*]). Indeed, hemodynamics and in particular shear stress was shown to control endothelial cell morphology and influence the expression of many transcription factors (for example, for a link between flow, mechanotransduction, actin, transcription see *Nakajima et al., 2017*).

## Biomechanics of the apical constriction compared to other morphogenetic processes

The constriction of the apical/luminal side of EHT cells, like many apical cell constriction events described so far, should depend on the activity of actomyosin recruited at the junctional circumferential belt. The membrane interfaces that are the less enriched in actomyosin (oriented along the antero-posterior axis), are the ones that shrink the most, supporting the idea that the constriction is also coupled with their consumption, possibly by endocytosis.

EHT cell apical constriction exhibits peculiar biomechanical characteristics in comparison to other apical constriction mechanisms described so far. It is pulsatile, with successive phases of contraction and of stabilization, each one with a time-length varying stochastically (see *Figure 1—figure supplement 1D* and *Figure 2—figure supplement 1*). The pulsatility relies on variations of the apical closing speed that oscillates (owing to acceleration and deceleration cycles), with more or less large amplitudes (with larger amplitudes during the constriction than the stabilization phase) and with periods of approximately 40 min on average. This average period value is longer than in the most extensively studied and quantified processes taking place during development, such as, for example, the apical-medial contraction of cells during ventral mesoderm invagination in *Drosophila melanogaster* embryos (see the seminal article [*Martin et al., 2009*]), or the apical-medial and apical-

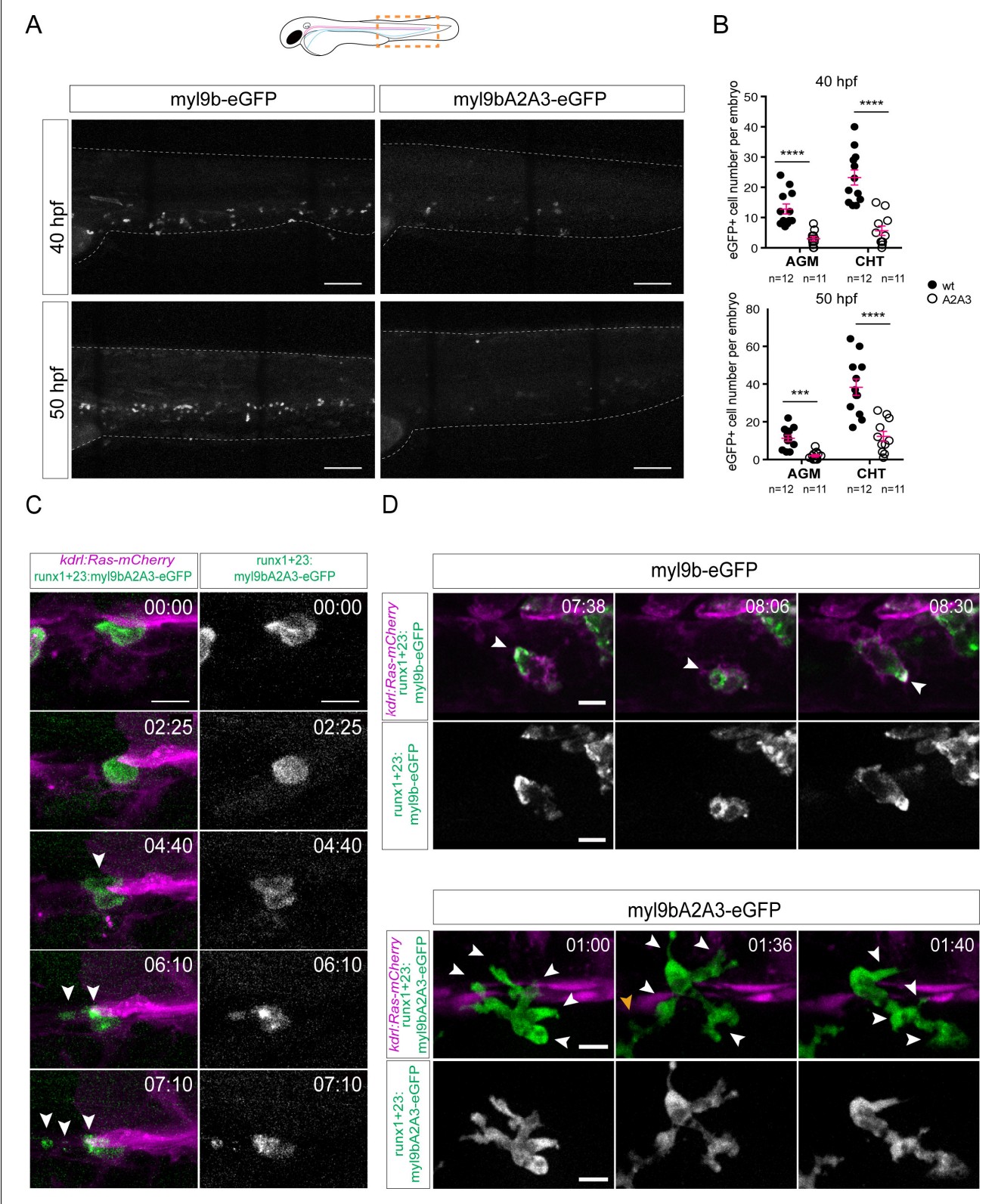

**Figure 9.** The amino-terminal phosphorylation site on Myl9b is essential for definitive hematopoiesis. (**A**) Maximum projection of Z-planes obtained from 40 hpf and 50 hpf embryos expressing either Myl9b-eGFP or Myl9bA2A3-eGFP under the control of the runx'1 + 23' enhancer that allows expression in hematopoietic stem cells (see also *Figure 9—figure supplement 1*). White dotted lines delineate the embryo. Scale bars, 100 μm. (**B**) Quantification of the number of Myl9b-eGFP or Myl9bA2A3-eGFP positive cells in the AGM and CHT at 40 hpf or 50 hpf. Error bars represent mean

*Figure 9 continued on next page*

*Figure 9 continued*

values ± SEM. Unpaired t test (***) p<0.001; (****) p<0.0001 (n = 11 and n = 12 embryos; the data are representative of two independent experiments, both performed at 40 hpf and 50 hpf). (C–D) Maximum projection of Z-planes extracted from spinning-disk confocal TL sequences performed on 40 hpf *Tg(kdrl:Ras-mCherry)* embryos expressing the indicated forms of Myl9b-eGFP. Scale bars, 10 µm. Time is indicated in hr:min. (C) White arrowheads, cell debris resulting from bursting (see *Figure 9—video 2*). Similar phenotype was observed in three independent experiments. (D) White arrowheads, Myl9b-eGFP accumulation at the rear of a migrating cell (top panel) or at multiple cell extensions lacking Myl9bA2A3-eGFP accumulation (bottom panel). Similar phenotype was observed in four independent experiments. Yellow arrowhead, another migrating cell within the imaging field (see *Figure 9—video 1*).

DOI: https://doi.org/10.7554/eLife.37355.031

The following video and figure supplement are available for figure 9:

**Figure supplement 1.** The runx'1 + 23' enhancer allows expression of Myl9 in hematopoietic stem cells (see also text).

DOI: https://doi.org/10.7554/eLife.37355.032

**Figure 9—video 1.** Phenotype of Myl9b phosphorylation mutant −1.

DOI: https://doi.org/10.7554/eLife.37355.033

**Figure 9—video 2.** Phenotype of Myl9b phosphorylation mutant −2.

DOI: https://doi.org/10.7554/eLife.37355.034

junctional mediated contractions during neuroblasts ingression (see [*An et al., 2017*; *Simões et al., 2017*]); the latter being more comparable to the EHT process since a single cell extrudes from an epithelium, the ectoderm. In these two processes, for which the duration of the apical constriction takes approximatey 6 min and 30 min, respectively, the period of each pulsatile cycle is in the minute range (with, for example, an average of 3.2 constriction pulses over 6 min for ventral cells (interspersed with inter-periodic intervals of similar values) and of 28.8 constriction pulses over 30 min for neuroblasts (value extrapolated from *Simões et al., 2017*)). Finally, for the EHT, the inter-periodic intervals of the contraction phase take approximately 21 min on average (which we referred to as pseudo-plateaus, see *Figure 2—figure supplement 1*). Currently, the reason for such a discrepancy in the periods of pulsatile cycles of the apical constriction is not known. However, in comparison to the time scale of apical contraction in the neuroblast ingression process that also leads to cell extrusion (lasting for approximately 30 min, as mentioned already), the timescale of the EHT process per se is relatively long (average duration of approximately 10 hr). This suggests that the specific parameters of the oscillations of the apical closing speed for EHT cells that impact on contraction efficiencies (in particular for periods and amplitudes) correlate with the time scale of the whole process, suggesting that they may be controlled at the mesoscopic level, perhaps involving gene regulatory networks controlling cyclic expression of genes (such as in the case of ultradian oscillations that have periods in the time-range of the hour, see *Isomura and Kageyama, 2014*). Indeed, EHT cells are integrated into a developing lumenized tissue that is submitted to a series of mechanical tensions owing to aortic morphological maturation including (i) the adaptation of the endothelial wall to EHT cell emergence and the significant reduction of the aortic diameter with time (*Isogai et al., 2001*; *Kissa and Herbomel, 2010*; *Sugden et al., 2017*), (ii) the extensive modification of the structure of the sub-aortic space (with the modification of the basal lamina contacting EHT cells and transducing forces via integrins; see the incidence of basal lamina on periods of oscillations [*Gorfinkiel, 2016*; *Koride et al., 2014*]) and, last but not least, (iii) complex hemodynamic forces. In such environment, the high mechanical tensions might tune the downstream molecular events controlling actin-driven directional forces and actomyosin recruitment and contractility (for the relevance of high-order information, at the tissue level, on the biomechanics and role of contractile pulses and the theoretical description of pulsatile/oscillatory behaviour, see [*Levayer and Lecuit, 2012*]).

## The regulation of actomyosin contraction

In this study, we do show that the contraction of circumferential actomyosin accompanies the constriction of EHT cells; however, we do not show a correlation between apical pulses and oscillatory recruitment of actomyosin components in that specific region as performed in other studies, essentially for technical reasons (the weakness of signals of myosin RLCs fused to fluorescent reporters in all the transgenic lines established is too close to background to allow extraction of oscillatory signals from fluctuating noise). Hence, we can only speculate that the peculiarities of the biomechanical characteristics of apical constriction discussed above rely on the regulation of actomyosin

recruitment and contraction. However, interfering with *Myl9b* function by mutagenesis of its PKC phosphorylation site impairs hematopoiesis, which is consistent with the requirement for actomyosin activity during the EHT and points to the importance of a particular regulatory pathway. Indeed, the N-terminal phosphorylation site in *Myl9b*, shared with most other myosin RLCs, was shown to be the substrate of PKC (*Ikebe et al., 1987*) and, upon phosphorylation, to prevent the activating phosphorylation of a downstream site (the MLCK site, the most characterized phosphorylation site of RLCs, see [*Turbedsky et al., 1997*]). Hence, this reinforces the possibility that a major regulatory axis of the EHT process relies on controlling actomyosin activity, by regulating a complex interplay between activating and inhibitory kinases. In this scenario, there exists the possibility that *Myl9b*, via the phosphorylation of its N-terminus, contributes to the tuning of the mechanical parameters of the contraction. Finally, there may be additional complexity to the system, for another RLC, *Myl9a*, is expressed in the aorta and may be co-expressed with *Myl9b* in EHT cells. *Myl9a* is a paralog that appears to be more restricted to the zebrafish and lacks the N-terminal, PKC phosphorylation site. Since the knock-down of *myl9a*, as is the case for *myl9b*, impairs definitive hematopoiesis, both isoforms may be involved in the EHT, with a fine tuning of their molecular ratio.

In conclusion, our study reveals key biochemical and biomechanical features of hematopoietic stem cell emergence in the zebrafish embryo and strengthens the significance of the mechanical constraints imposed by the environment and more specifically the blood flow. This work paves the way for future studies on this very unique process, including studying the contribution of forces exerted by the adjoining endothelial cells. Overall, it invites to biophysical modelling to point to the most relevant biomechanical features of the process.

# Materials and methods

**Key resources table**

| Reagent type (species) or resource | Designation | Source or reference | Identifiers | Additional information |
|---|---|---|---|---|
| Strain, strain background (Zebrafish) | Zebrafish: AB | Zebrafish International Resource Center (ZIRC) | ZFIN: ZDB-GENO-960809–7 | |
| Strain, strain background (Zebrafish) | Zebrafish Tg(kdrl:HsHRAS -mCherry): s916Tg | *Chi et al. (2008)* | ZFIN ID: ZDB-ALT-090506–2 | referred to as Tg(kdrl:Ras-mCherry) |
| Strain, strain background (Zebrafish) | Zebrafish: Tg(fli1a:LIFEACT-eGFP): zf495Tg | *Phng et al. (2013)* | ZFIN ID: ZDB-ALT-140610–8 | |
| Strain, strain background (Zebrafish) | Zebrafish: Tg(5xUAS:RFP): nkuasrfp1aTg | *Asakawa et al. (2008)* | ZFIN ID: ZDB-ALT-080528–2 | |
| Strain, strain background (Zebrafish) | Zebrafish: Tg(fli1ep:Gal4ff): ubs3Tg | *Zygmunt et al. (2011)* | ZFIN ID: ZDB-ALT-120113–6 | |
| Strain, strain background (Zebrafish) | Zebrafish: Tg(UAS:EGFP-ZO1): ubs5Tg | *Herwig et al. (2011)* | ZFIN ID: ZDB-ALT-120113–7 | |
| Strain, strain background (Zebrafish) | Zebrafish: Tg(kdrl:EGFP): s843Tg | *Jin et al. (2005)* | ZFIN ID: ZDB-ALT-050916–14 | |
| Strain, strain background (Zebrafish) | Zebrafish: Tg(kdrl:gal4;UAS:RFP) | NA | NA | |
| Strain, strain background (Zebrafish) | Zebrafish Tg(−6.0itga2b:EGFP): la2Tg | *Lin et al., 2005* Traver | ZFIN ID: ZDB-ALT-051223–4 | referred to as Tg(CD41:eGFP) |

*Continued on next page*

*Continued*

| Reagent type (species) or resource | Designation | Source or reference | Identifiers | Additional information |
|---|---|---|---|---|
| Strain, strain background (Zebrafish) | Zebrafish Tg(kdrl:myl9a-eGFP): ip5Tg | This Paper | N/A | |
| Strain, strain background (Zebrafish) | Zebrafish Tg(kdrl:myl9b-eGFP): ip6Tg | This paper | N/A | |
| Antibody | Sheep Anti-Digoxigenin Fab fragments, POD Conjugated | Roche | Cat# 11207733910; RRID: AB_514500 | 1:5000 |
| Recombinant DNA reagent | Transposase pCS-zT2TP | *Suster et al. (2011)* | N/A | |
| Recombinant DNA reagent | pUAS:eGFP-hs-ZO1 | *Herwig et al. (2011)* | N/A | |
| Recombinant DNA reagent | pEGFP-C1 | Clontech | N/A | |
| Recombinant DNA reagent | pmKate2-f-mem | Evrogen | Cat#: FP186 | |
| Recombinant DNA reagent | pG1-flk1-MCS-tol2 | *Jin et al. (2005)* | Addgene plasmid # 26436 | |
| Recombinant DNA reagent | PG1-flk-MCS-tol2-kdrl-myl9a-eGFP | This paper | N/A | |
| Recombinant DNA reagent | PG1-flk-MCS-tol2-kdrl-myl9b-eGFP | This paper | N/A | |
| Recombinant DNA reagent | p5E-Runx1 + 23-betaglobin | *Tamplin et al. (2015)* | Addgene plasmid # 69602 | |
| Recombinant DNA reagent | pTol2-Runx1 + 23×1-betaglobin:eGFP | This paper | N/A | |
| Recombinant DNA reagent | pTol2-Runx1 + 23×1-betaglobin-myl9b-eGFP | This paper | N/A | |
| Recombinant DNA reagent | pTol2-Runx1 + 23×1-betaglobin-myl9bA2A3-eGFP | This paper | N/A | |
| Recombinant DNA reagent | pTol2-Runx1 + 23×1-betaglobin-myl9a-mKate2 | This paper | N/A | |
| Chemical compound, drug | Tricaine | Sigma Aldrich | Cat#: A5040 | |
| Chemical compound, drug | Methylen Blue | Sigma Aldrich | Cat#: M4159 | |
| Chemical compound, drug | N-Phenylthiourea (PTU) | Sigma Aldrich | Cat#: P7629 | |
| Chemical compound, drug | Low Melting Point, Analytical Grade | Promega | Cat#: V2111 | |
| Chemical compound, drug | Formaldehyde (FA) | Polysciences | Cat#:04018–1 | |
| Chemical compound, drug | Proteinase K | Ambion | Cat#: AM2546 | |
| Chemical compound, drug | Digoxigenin labeled nucleotides | Jena Bioscence | Cat#: NU-803-DIGX | |
| Chemical compound, drug | Wester Blotting Reagent (WBR) | Roche | Cat#: 11921673001 | |
| Chemical compound, drug | NBT | Sigma Aldrich | Cat#: N6639 | |
| Chemical compound, drug | BCIP | Sigma Aldrich | Cat#: B-8503 | |

*Continued*

| Reagent type (species) or resource | Designation | Source or reference | Identifiers | Additional information |
|---|---|---|---|---|
| Chemical compound, drug | Tissue Freezing Medium, Blue | Electron Microscopy Sciences | Cat#: 72592-B | |
| Chemical compound, drug | Aqua-Poly/Mount | Biovalley | Cat#: 18606–5 | |
| Chemical compound, drug | Immersol W 2010, oil | Zeiss | Cat#: 444969-0000-000 | |
| Chemical compound, drug | AquaSound Clear ultrasound gel | Free-Med | Cat#: FU00071 | |
| Commercial assay or kit | Cloning Gibson Assembly | NEB | Cat#: E2611S | |
| Commercial assay or kit | cDNA synthesis for cloning Super Script III | Thermo Fisher | Cat#: 8080093 | |
| Commercial assay or kit | Expand High Fidelity polymerase | Sigma Aldrich | Cat#:11732641001 | |
| Commercial assay or kit | RNA synthesis mMESSAGE mMACHINE SP6 Transcription | Ambion | Cat#: AM1340 | |
| Commercial assay or kit | Poly(A) Tailing Kit | Ambion | Cat#: AM1350 | |
| Commercial assay or kit | DNA purification for injection NucleoBond Xtra Midi EF | Macherey Nagel | Cat#: 740420.10 | |
| Commercial assay or kit | T7 RNA polymerase | Promega | Cat#: P2075 | |
| Commercial assay or kit | RNA extraction RNeasy Plus Mini Kit | QIagen | Cat#: 74134 | |
| Commercial assay or kit | cDNA synthesis for qPCR M-MLV Reverse Transcriptase | Invitrogen | Cat#: 28025013 | |
| Commercial assay or kit | RNAse Out | Invitrogen | Cat#: 10777019 | |
| Commercial assay or kit | qPCR Takyon Rox SYBR Master mix blue dTTP | Eurogenetec | Cat#: UF-RSMT-B0701 | |
| Commercial assay or kit | NucAwayTM d'AmbionTM | Invitrogen | Cat#: AM10070 | |
| Software, algorithm | Acquisition-Imaging analysis-LAS X | Leica | http://www.Leica-microsystems.com | |
| Software, algorithm | Acquisition-Volocity | Perkin Elmer | http://www.perkinelmer.com/ | |
| Software, algorithm | Imaging analysis-Imaris 8.4.1 | Bitplane | http://www.bitplane.com/ | |
| Software, algorithm | Imaging analysis-Matlab R2017a/R2017b | MathWorks | https://fr.mathworks.com | |
| Software, algorithm | Image Analysis-Prism 6 | Graph Pad | https://www.graphpad.com/ | |
| Software, algorithm | Image Analysis-Fiji | NIH | https://imagej.net/Fiji | |
| Software, algorithm | Image Analysis TrackMate plugin for Fiji | *Tinevez et al. (2017)* | https://imagej.net/TrackMate | |
| Software, algorithm | Image Analysis Icy | *de Chaumont et al., 2012* | http://icy.bioimageanalysis.org/ | |
| Software, algorithm | Image Analysis TubeSkinner plugin for Icy | This paper | This Paper | |

*Continued on next page*

*Continued*

| Reagent type (species) or resource | Designation | Source or reference | Identifiers | Additional information |
|---|---|---|---|---|
| Software, algorithm | Figures-Photoshop CC 2017.1.1 | Adobe | http://www.adobe.com/cn/ | |
| Software, algorithm | Figures-Illustrator CC 2017.1.1 | Adobe | http://www.adobe.com/cn/ | |
| Recombinant DNA reagent | Morpholino Standard Control | Gene Tools | N/A | CCTCTTACCTCAGTTACAATTTATA |
| Recombinant DNA reagent | Morpholino sih | *Sehnert et al. (2002)* | ZDB-MRPHLNO-060317–4 | CATGTTTGCTCTGATCTGACACGCA |
| Recombinant DNA reagent | Morpholino Myl9aspe2i2 | Gene Tools; This paper | N/A | ATTCAGCTTGTATATCTCTCACCCA |
| Recombinant DNA reagent | Morpholino Myl9aspi3e4 | Gene Tools; This paper | N/A | CCCTGGTACAAACACACCGCAGATT |
| Recombinant DNA reagent | Morpholino Myl9bspe2i2 | Gene Tools; This paper | N/A | CTATTATTTCACCCAGACTCACCCA |
| Recombinant DNA reagent | Morpholino Myl9bspe3i3 | Gene Tools; This paper | N/A | TTGTTGTTGTTTTTACCAGATCCCT |
| Recombinant DNA reagent | Oligo: Rescue-SP6-myl9b forward | This paper | N/A | atttaggtgacactatagaagngATGTCCAGCAAAAGAGCAAAGGGG |
| Recombinant DNA reagent | Oligo: Rescue-myl9b reverse | This paper | N/A | CTACATGTCGTCTTTGTCTTTGGCTCCG |
| Recombinant DNA reagent | Oligo: qPCR myl9a forward | This paper | N/A | CACCTGCTTCGATGAGGACGCCA |
| Recombinant DNA reagent | Oligo: qPCR myl9a reverse | This paper | N/A | CCGAGATCTGCTGACCACTATGGGCGATC |
| Recombinant DNA reagent | Oligo: qPCR myl9b forward | This paper | N/A | GCCTGCTTCGATGAGGAGGGATC |
| Recombinant DNA reagent | Oligo: qPCR myl9b reverse | This Paper | N/A | CTGACCACCATGGGAGACC |
| Recombinant DNA reagent | Oligo: qPCR splicemyl9asp22 forward | This paper | N/A | CACGTCCAATGTCTTCGCCA |
| Recombinant DNA reagent | Oligo: qPCR splicemyl9asp22 reverse | This paper | N/A | TGGGTTCTTCCCCAGAGAGG |
| Recombinant DNA reagent | Oligo: qPCR splicemyl9asp34 forward | This Paper | N/A | GGACGCCACTGGGTTCATCC |
| Recombinant DNA reagent | Oligo: qPCR splicemyl9asp34 reverse | This paper | N/A | GTTTCCTTTCTTGTCGATGGGCGC |
| Recombinant DNA reagent | Oligo: qPCR splicemyl9bsp22 forward | This paper | N/A | GCATCTTTGGGTAAGAACCCGTCTG |
| Recombinant DNA reagent | Oligo: qPCR splicemyl9bsp22 reverse | This paper | N/A | TCAGCCGCTCTCCAAACATG |
| Recombinant DNA reagent | Oligo: qPCR splicemyl9bsp33 forward | This paper | N/A | GAGGAGGGATCTGGTTTCATCC |
| Recombinant DNA reagent | Oligo: qPCR splicemyl9bsp33 reverse | This paper | N/A | CCCTTCTTGTCAATGGGAGCC |

*Continued on next page*

*Continued*

| Reagent type (species) or resource | Designation | Source or reference | Identifiers | Additional information |
|---|---|---|---|---|
| Recombinant DNA reagent | Oligo: Probe myl9a forward | This paper | N/A | GATAAGGAGGATCTGCATGA CATGCTCG |
| Recombinant DNA reagent | Oligo: Probe T7 myl9a reverse | This paper | N/A | gaaattaatacgactcactatagg GTGAAGCGATCGCCCATAGTGG |
| Recombinant DNA reagent | Oligo: Probe myl9b forward | This paper | N/A | GCACGATATGCTAGCATCTTTGGG |
| Recombinant DNA reagent | Oligo: Probe T7-myl9b reverse | This paper | N/A | gaaattaatacgactcactataggg CATGGTGGTCAGCAGCTCCC |
| Recombinant DNA reagent | Oligo for Cloning: PG1-flk-MCS-tol2-myl9a-eGFP-myl9a-for | This paper | N/A | TATTTTAACAGACAAGGGCGATGT CTGCAGCCAAACGCGCCAAAGGAAAG |
| Recombinant DNA reagent | Oligo for Cloning: PG1-flk-MCS-tol2-myl9a-eGFP-myl9a-rev | This paper | N/A | CCTTGCTCACCATGTCGTCCTTG TCTTTGGCTCCGTGCTTTAG |
| Recombinant DNA reagent | Oligo for Cloning: PG1-flk-MCS-tol2-myl9a-eGFP-eGFP-for | This paper | N/A | GGACGACATGGTGAGCAAGGGCG AGGAGCTGTTCACCGGG |
| Recombinant DNA reagent | Oligo for Cloning: PG1-flk-MCS-tol2-myl9a-eGFP-eGFP-rev | This paper | N/A | CGATATCCTCGAGGGTACCGTTATCTA GATCCGGTGGATCCCGGGCCCGC |
| Recombinant DNA reagent | Oligo for Cloning: PG1-flk-MCS-tol2-myl9b-eGFP-myl9b-for | This paper | N/A | GACGTTATTTTAACAGACAAGGGCG ATGTCCAGCAAAAGAGCAAAGGGG AAGACC |
| Recombinant DNA reagent | Oligo for Cloning: PG1-flk-MCS-tol2-myl9b-eGFP-myl9b-rev | This paper | N/A | CGCCCTTGCTCACCATGTCGTCTTTGT CTTTGGCTCCGTGTTT |
| Recombinant DNA reagent | Oligo for Cloning: PG1-flk-MCS-tol2-myl9b-eGFP-eGFP-for | This paper | N/A | CAAAGACGACATGGTGAGCAAGGGCG AGGAGCTGTTCACCGGG |
| Recombinant DNA reagent | Oligo for Cloning: PG1-flk-MCS-tol2-myl9b-eGFP-eGFP-rev | This paper | N/A | GGATCCGATATCCTCGAGGGTACCG TTATCTAGATCCGGTGGATCCCG GGCCCGC |
| Recombinant DNA reagent | Oligo for Cloning: pTol2-Runx1 + 23-betaglobin-myl9a-mKate2-myl9a-for | This paper | N/A | ATCCCGCGGTGGAGCTCCAGAATTCAT GTCTGCAGCCAAACGCGCCAAAGG |
| Recombinant DNA reagent | Oligo for Cloning: pTol2-Runx1 + 23-betaglobin-myl9a-mKate2-myl9a-rev | This paper | N/A | GCATGTTCTCCTTAATCAGCTCGCTC ACCATGTCGTCCTTGTCTTTGGCTCC |
| Recombinant DNA reagent | Oligo for Cloning: pTol2-Runx1 + 23-betaglobin-myl9a-mKate2-mKate2-for | This paper | N/A | GAGCCAAAGACAAGGACGACATGGT GAGCGAGCTGATTAAGGAGAACATGC |
| Recombinant DNA reagent | Oligo for Cloning: pTol2-Runx1 + 23-betaglobin-myl9a-mKate2-mKate2-rev | This paper | N/A | GGATCCGATATCCTCGAGGGTACCGTC ATCTGTGCCCCAGTTTGCTAGGG |
| Recombinant DNA reagent | Oligo for Cloning: pTol2-Runx1 + 23×1-betaglobin-myl9b-eGFP-myl9a-for | This paper | N/A | CAGACATCCCGCGGTGGAGCTCC AGGTCGCCACCATGTCCAGCAAAAG AGCAAAGGGGAAGACCACCAAG |
| Recombinant DNA reagent | Oligo for Cloning: pTol2-Runx1 + 23×1-betaglobin-myl9b-eGFP-myl9a-rev | This paper | N/A | CGCCCTTGCTCACCATggtggcgacGAA TTCCATGTCGTCTTTGTCTTTGG CTCCGTG |
| Recombinant DNA reagent | Oligos for mutagenesis: pTol2-Runx1 + 23×1-betaglobin-myl9b-eGFP-myl9aA2A3-for | This paper | N/A | CAGACATCCCGCGGTGGAGCTCCA GgtcgccaccAtggccgccaaaagagca aaggggaagaccaccaag |

*Continued on next page*

*Continued*

| Reagent type (species) or resource | Designation | Source or reference | Identifiers | Additional information |
|---|---|---|---|---|
| Recombinant DNA reagent | Oligo for mutagenesis: pTol2-Runx1 + 23×1 betaglobin- myl9b-eGFP-<br><br>myl9aA2A3-rev | This paper | N/A | CGCCCTTGCTCACCATggtggcgac GAATTCcatgtcgtctttgtctttg gctccgtg |

## Contact for reagent and resource sharing

Further information and requests for resources and reagents should be directed to and will be fulfilled by the corresponding Authors, Anne A. Schmidt (anne.schmidt@pasteur.fr) and Philippe Herbomel (philippe.herbomel@pasteur.fr).

## Zebrafish husbandry

Zebrafish (*Danio rerio*) of the AB background and transgenic fish carrying the following transgenes *Tg(kdrl:HsHRAS-mCherry)*[s916] (*Chi et al., 2008*) also referred to as *Tg(kdrl:Ras-mCherry)*, *Tg(fli1a:Lifeact-eGFP)*[zf495] (*Phng et al., 2013*), *Tg(fli1ep:Gal4ff)*[ubs3] (*Zygmunt et al., 2011*), *Tg (5xUAS:RFP)*[nkuasrfp1a] (*Asakawa et al., 2008*), *Tg(UAS:eGFP-ZO1)*[ubs5] (*Herwig et al., 2011*), *Tg(kdrl: eGFP)*[s843] (*Jin et al., 2005*), *Tg(−6.0itga2b:eGFP)*[la2] (*Lin et al., 2005*) also referred to as *Tg(CD41: eGFP)*, *Tg(kdrl:Gal4; UAS:RFP)*, *Tg(kdrl:Myl9a-eGFP)*[ip5] and *Tg(kdrl:Myl9b-eGFP)*[ip6] (this study) were raised and staged as previously described (*Kimmel et al., 1995*). Fish were maintained on a 14 hr light/10 hr dark cycle, and embryos were collected and raised at 28.5 or 24°C in N-Phenylthiourea (PTU, Sigma Aldrich, Cat#: P7629)/Volvic source water (0.003%, final) to prevent pigmentation implemented with 280 µg/L methylene blue (Sigma Aldrich, Cat#: M4159). The embryos used were of early developmental stages (26–50 hpf), precluding sex determination of the animals. The general fish maintenance at the Institute follows the regulations of the 2010/63 UE European directives and is supervised by the veterinarian office of Myriam Mattei.

## Transient and stable transgenesis

All constructs were generated using the Gibson Assembly method (NEB, Cat#: E2611S). The *myl9a* and *b* sequences were amplified from a pool of cDNAs made using 24 to 48 hpf whole zebrafish embryos (see Key Resources Table for the primers used, Super Script III kit, Cat#: 8080093 and Expand High Fidelity polymerase, Sigma Aldrich, Cat#:11732641001). The *kdrl* promoter (*Flk*) (*Jin et al., 2005*) was used to drive endothelial expression of Myl9a and Myl9b fused at their carboxy-terminus, with eGFP (Clontech) or mKate2 (Evrogen). The murine runx'1 + 23' enhancer (*Tamplin et al., 2015*) coupled with the murine betaglobin was used to drive hemogenic expression. Final constructs were purified using the NucleoBond Xtra Midi endotoxin free kit (Macherey Nagel, Cat#: 740420.10) and 25 ng/µl were co-injected into one-cell stage zebrafish embryos along with 25 ng/µl *tol2* transposase mRNA transcribed from linearized pCS-zT2TP (*Suster et al., 2011*) plasmid using the mMESSAGE mMACHINE SP6 kit (Ambion, Cat#: AM1340) (*Kawakami, 2007*; *Kotani et al., 2006*). Embryos were screened for fluorescence between 24 hpf and 48 hpf, selected for imaging or raised to adulthood for stable transgenesis. Founders were isolated by screening for fluorescence. For stable transgenesis, the following plasmids have been injected: pG1-flk-MCS-tol2-Myl9a-eGFP or pG1-flk-MCS-tol2-Myl9b-eGFP. For transient transgenesis, the following plasmids have been used: pTol2-Runx1+23-1-betaglobin-myl9b-eGFP, pTol2-Runx1 + 23-1-betaglobin-myl9bA2A3-eGFP, pTol2-Runx1 + 23-1-betaglobin-Myl9a-mKate2, pUAS:eGFP-hs-ZO1.

## Mutagenesis

The Myl9b phosphorylation mutant form was generated by site directed mutagenesis using a sense 5-prime primer containing three nucleotide substitutions and PCR amplification using the wild-type *myl9b* cDNA as template. The sense primer also contained a Kozak sequence. Sense and anti-sense primers contained extensions for cloning the PCR product into the pTol2-Runx1+23-1-beta-globin: eGFP using the Gibson Assembly method. The double mutation was verified by sequencing.

Sense mutagenesis primer (with the substituted nucleotides in underlined bold):

CAGACATCCCGCGGTGGAGCTCCAGGTCGCCACCATG
**G**CC**GC**CAAAAGAGCAAAGGGGAAGACCACCAAG
Reverse primer:
CGCCCTTGCTCACCATGGTGGCGACGAATTCCATGTCGTCTTTGTCTTTGGCTCCGTG

## Morpholino microinjections

Splice blocking morpholinos (MOs, see Key Resources Table for sequences), myl9aspe2i2, myl9as-pi3e4, myl9bspe2i2, myl9bspe3i3, and as well as the *sih* Tnnt2 translation start codon and flanking 5-prime sequence MO (*Sehnert et al., 2002*) were prepared in stock solutions at 2 mM in ddH$_2$O. 1.5 (sih MO) or 4–12 ng (*myl9a* and *myl9b* MOs) were injected into one-cell stage zebrafish embryos. The same amount of MO targeting a mutated form of the human beta-globin pre-mRNA was used as control (Standard Control from GeneTools). Splice blocking efficiency was confirmed for each *myl9* MO using qRT-PCR assays (see Key Resources Table for sequences) and the resulting pheno-types were assessed by manual counting of CD41:eGFP-positive cells in the AGM and the CHT of 40 and 50 hpf embryos. Results obtained using the MOs myl9aspe2i2 (MO *myl9a*, 12 ng) and the MO myl9bspe3i3 (MO *myl9b*, 12 ng) are presented in *Figure 8*. For *sih* morphants, embryos that were imaged were checked for absence of heart beating 24 hr after injection as well as before and after experimental analysis.

## Rescue experiment

Capped and poly-adenylated full-length *myl9b* mRNAs were synthesized using the mMESSAGE mMACHINE kit (Ambion, Cat#: AM1340) and the Poly(A) Tailling kit (Ambion, Cat#: AM1350) from *myl9b* cDNA amplified by PCR from our pG1-flk-MCS-tol2-Myl9b-eGFP expression vector. 100 or 200 pg of *myl9b* mRNA were microinjected into one-cell stage *Tg(CD41:eGFP)* embryos either alone or in combination with 6 ng or 12 ng of myl9bspe3i3 morpholino. The resulting phenotypes were assessed by manual counting of CD41:eGFP-positive cells in the AGM and the CHT of 48 and 50 hpf embryos.

## Whole-mount in situ hybridization and cryosections

Whole-mount chromogenic in situ hybridization was performed as described previously (*Thisse and Thisse, 2014*). Briefly, embryos were hand dechorionated, fixed in 4% FA (Polysciences, Cat#: 040181) overnight at 4°C and stored in 100% MeOH at −20°C. Embryos were permeabilized using 10 µg/ml Proteinase K (Ambion, Cat#: AM2546) in PBT (1x PBS + 0.1% Tween) for 5 min and post-fixed in 4% FA for 20 min. Hybridization was performed using 200 ng Digoxigenin-labeled myl9a (RefSeq: NM_001006027) and myl9b (RefSeq: NM_213212) probes synthetized using the man-ufacturer recommendation (T7 RNA polymerase, Promega, Cat#: P2075, DIG-nucleotides, Jena Bio-scence, Cat#: NU-803-DIGX). Detection of the probes was performed by incubating the embryos with an anti-Digoxigenin antibody coupled to alkaline phosphase (Roche, Cat#: 11207733910) diluted 1:5000 in blocking solution 1X WBR (Roche, Cat#: 11921673001). The staining reaction was performed using NBT/BCIP (Sigma, Cat#: N6639, Cat#: B-8503) in NTMT buffer (100 mM Tris HCl pH 9.5, 50 mM MgCl2, 100 mM NaCl, 0.1% Tween). Subsequently to the revelation, embryos were washed in PBT and incubated in 30% sucrose for 2 days at 4°C and with agitation before embedding in tissue freezing medium blue (Electron Microscopy Sciences, Cat#: 72592-B) and frozen in liquid nitrogen. Transverse sectioning was performed at 5 µm using a CM 3050 cryostat (Leica) and slices were dried for 2 hr before mounted with AquaPoly Mount medium (Biovalley, Cat#: 18606–5). Images were captured on an Olympus vs120 slide scanner using an ORCA Flash4.0 V2 CMOS cam-era (Hamamatsu) and a UPLSAPO 20X objective (Olympus).

## Quantitative real-time PCR

To determine the splice-blocking efficiency of *myl9a* and *myl9b* MOs, cDNA was extracted from 20 whole 48 hpf embryos and qRT-PCR oligos were designed to hybridize on the joined sequential exons initially surrounding the splicing sites targeted by the MOs (see Key Resources Table for sequences). One-cell-stage embryos were injected with 4–8 ng of *myl9a* or *myl9b* or Standard con-trol MOs (see Key Resources Table for sequences) and raised until 48 hpf. Total mRNA was extracted using the RNeasy Plus Mini Kit (Qiagen, Cat#: 74134) as indicated by the manufacturer

and the concentration was obtained by nanodrop quantification (Nanodrop1000, ThermoFisher). Equivalent amounts of RNA (300 ng) were used as template for cDNA synthesis, which was performed using the M-MLV Reverse Transcriptase kit (Invitrogen, Cat#: 28025013). qRT-PCR was performed using Takyon Rox SYBR Master mix blue dTTP kit (Eurogentec, Cat#: UF-RSMT-B0701), 15 ng of cDNA and 10pM of each primer. qRT-PCR was carried out for three biological replicates with measurements taken from three technical replicates on an Applied Biosystems 7300 Real Time PCR system (Thermofisher). The relative expressions of *myl9a* and *9b* were determined after normalization to zebrafish elongation factor 1a (*eif1α*). The conditions from the samples injected with the *myl9* MOs were compared to the Control ones using the delta-delta-Ct method. To determine the potential off target effect of *myl9a* MOs on *myl9b* expression and of *myl9b* MOs on *myl9a* expression (see Key Resources Table for oligo sequences), RNA extraction was performed as described above; cDNA synthesis was performed using the SuperScript III kit (ThermoFisher, Cat#: 8080093), equivalent amount of RNA (200ng) and qRT-PCR were performed as described above. The graph presented in *Figure 8—figure supplement 1D* corresponds to two independent experiments carried out in triplicates. Due to the low n number (n = 2), this graph indicates a tendency.

## Embryo mounting for confocal imaging

Embryos were hand dechorionated and anesthesized using tricaine (Sigma Aldrich, Cat#: A5040). Then, they were mounted in lateral position and embedded in 1% low melting agarose (Promega, Cat#: V2111) in a glass-bottom μ-Dish (Ibidi, Cat#: 81158). To avoid movements and pigmentation during image acquisitions, 1x tricaine/1 x PTU Volvic water was incorporated to the low melting agarose and, after solidification, added on top of it.

## In vivo spinning-disk confocal imaging

Embryos were imaged on an inverted microscope (UltraVIEW VOX, Perkin Elmer) equipped with a Yokogawa CSU-X1 spinning-disk, an ImagEM-X2 camera (Hamamatsu), 488 nm and 561 nm laser lines for excitation, a Zeiss LD C-Apochromat 40x corrM27 water objective (NA 1.1, WD 0.62 mm at cover glass 0.17 μm) and the Volocity (Perkin Elmer, http://www.perkinelmer.com/) acquisition software. Optical Z planes were spaced by 0.6 μm, the power of the lasers (around 150 mW) and exposure time (between 50 ms and 200 ms) were adjusted depending on the fluorophore and the transgenic line used. In all cases, acquisitions of a Z stacks containing the whole dorsal aorta was reduced to 1 min or less and spaced every 2 min. For TL sequences longer than 1 hr, samples were maintained at 28.5°C using an Okolab cage incubator and the objective was immersed in Immersol W 2010 oil (Zeiss, Cat#: 444969-0000-000).

## In vivo laser scanning confocal imaging

Embryos were imaged using an inverted microscope (Leica TCS SP8, CTR 6500), equipped with 488 nm and 561 nm diodes for excitation, a Hybrid Detector, a Leica HC PL APO CS2 40x water objective (NA 1.1, WD 0.65 mm at cover glass 0.17 μm) and the LAS X acquisition software (Leica, http://www.Leica-microsystems.com). Optical planes were spaced by 0.43 μm and Z stacks containing the whole dorsal aorta were acquired every 5-10 min. For long time-lapse imaging, samples were maintained at 28°C using a INU stage Top Incubator (Tokai Hit) and the objective was immersed in 1:1 AquaSound Clear (Free-med, Cat#: FU00071) ultrasound gel with water. For cell counting in live embryos, an HC PL APO CS2 20x IMM objective (NA 0.75, WD 0.67 mm) and a resonant scanner were used for high-speed imaging of the whole AGM or CHT. Subsequent maximum projection and manual counting was performed.

Some embryos were imaged with an upright microscope (Leica SPE, CTR 6000), equipped with 488 nm and 532 nm diodes for excitation, an immersion HCX APO LUV-I 40x water objective (NA 0.8, WD 3.3 mm) and the LAS AF version 2.6.0.7266 acquisition software (Leica). Optical planes were spaced by 0.6 μm and Z stacks containing the whole dorsal aorta were acquired every 5 min.

## AortaTracker algorithm for 2D projections

The algorithm developed for the projection of the aortic wall on a 2D plane, using fluorescent signals, was called AortaTracker. It is decomposed in two sections aimed at (1) the segmentation of the

aortic tube, (2) the unwrapping. To perform the segmentation of the tube, the algorithm first needs to be initialized by the user. To circumvent the variations in the aortic shape, movements, and signals coming from neighboring structures, a strong constrained surface is imposed to the aortic wall (this part is available as an Icy plugin (http://icy.bioimageanalysis.org) called TubeSkinner; [*de Chaumont et al., 2012*]). The aorta tube is rotated so that its axis becomes almost aligned with the Z-axis. On the first Z-section, the user then manually adjusts a circle fitting the inner part of the aorta. In such a section, the aorta is represented as a few numbers of bright spots representing the intersection of the plane with cellular membranes. The circle is then adjusted to maximize the sum of the fluorescent signals in a crown around its diameter, and to minimize the sum of intensity inside the inner circle of the crown. This procedure is repeated along the tube, effectively locating its central axis. Then for each circle, the intensity is collected along its perimeter as the maximal signal in the crown for a given angle. Finally, the unwrapper algorithm creates the image by transforming the point in the z-stack to the (theta, r) coordinate system considering the circle fit, making sure the final calibration is isotropic. No interpolation is performed on pixel intensity. The X-axis of the final image corresponds to the perimeter of the circles, and its Y-axis corresponds to the tube central axis.

## 2D-Maps representation
The obtained 2D-maps were then duplicated to merge the rims of unwrapped aorta and visualize the continuity of all cell contours.

## 3D-rendering
3D-rendering of live-imaging have been processed using the Imaris software (Bitplane, http://www.bitplane.com/).

## Manual A-P distance measurements
We considered the Antero-Posterior luminal distance (A-P distance), as the length between the most anterior and posterior luminal poles of the EHT undergoing cell, aligned with the A-P axis.

In the case of EHT cells emerging parallel to the A-P axis, we used the 'Straight Line' tool on Fiji software (NIH, https://imagej.net/Fiji) and measured the A-P distance (for each time points in *Figure 1—figure supplement 1*) on 2 or three consecutive Z-planes localized in the central part of the EHT cell (where the apparent A-P distance was the longest) and averaged.

For cells emerging at a small angle relative to the A-P axis, we measured the apparent A-P distance on the maximum projection along the Z-axis; when the angle was larger, we first performed a 3D-rotation on Imaris software so as to align the cell in the X-Y plane before doing the measurement.

## Semi-automatic A-P distance and closing speed measurements
In order to measure A-P luminal distances and dynamics (*Figure 2*), dense eGFP-ZO1 extremities were independently tracked using the Fiji plugin TrackMate (*Tinevez et al., 2017*, https://imagej.net/TrackMate) after maximum projection along the Z-axis and X,Y translation using the Fiji plugin 'Descriptor based series registration' to correct the X,Y drift (*Preibisch et al., 2010*). Tracks display and subsequent distance analysis were performed using the MATLAB software (MATLAB 2017a, , The MathWorks, https://fr.mathworks.com). To focus on long scale activity rather than short displacements, the distance between the two extremities was filtered twice using an unweighted sliding-average over 20 min. Following this process, we characterized the speed at which the A-P luminal distance decreases by automatically extracting the closing speed local minima and filtering them based on their prominence (>0.01 µm/min). For additional details see https://fr.mathworks.com/help/matlab/ref/islocalmin.html. Distance between successive minima was used to measure the acceleration-deceleration cycle duration distribution (for the algorithm and codes in MATLAB (version 2017b), see the link: https://github.com/sebherbert/EHT-analysis (*Herbert, 2018*; copy archived at https://github.com/elifesciences-publications/EHT-analysis).

## Tracks length
Distances in x were measured between the most anterior and posterior positions of the tracks.

## Morphometric analysis of aortic cells using 2D-maps

Cells were manually delimited using the 'Polygon Selections' tool on Fiji software and areas directly extracted from the measurement tool in Fiji. The length (parallel of the blood flow axis) and width (along the circumference of the aortic wall) were extracted from the rectangle that best fits the cell.

## Aortic cell types specification

To classify the aortic cells in one category or another (endothelial, potentially hemogenic or EHT cell, see *Figure 4*), we defined a series of criteria that take into account their respective length (along the antero-posterior axis), width (along the circumference of the aortic wall) and positioning in regard to the medio-lateral axis. By looking through the Z-planes and selecting the one for which the aorta is of the largest diameter in the dorso-ventral axis, we first determined the ventral floor axis and placed it on the 2D-map (see Floor on the maps). The roof (see Roof on the maps) was subsequently positioned at a distance corresponding to half the length of the unwrapped aorta (allowing to determine the surfaces of the frontside and backside). We then applied a relatively stringent cut-off and considered that the pseudo-hemogenic cell category did not extend beyond the medio-lateral axis of the aorta while spanning the ventral floor.

## Number of neighboring cells

Cells were considered as neighbors when at least a part of their lateral junctions were in direct contact.

## Statistical analysis

Statistical analysis was performed using Student unpaired t test with Welch's correction (does not assume similar SD) in GraphPad Prism six software (https://www.graphpad.com/). Graphs depict Mean ± SEM or SD or median alone and differences with p value $\leq 0.05$ (*) were considered statistically significant.

## Data and software availability

The AortaTracker algorithm for 2D projections is available as an Icy plugin (http://icy.bioimageanalysis.org).

## Acknowledgements

We are grateful to Romain Levayer for stimulating discussions, Denise Montell and Jocelyn McDonald for advice on myosin light chain mutagenesis, Emi Murayama for critical reading of the manuscript. We thank Leonard Zon for the p5E-R1+23 -1-beta-globin plasmid, Markus Affolter and Heinz Georg Belting for the pG1-flk1 and pUAS:eGFP-hs-ZO1 plasmids. This work was supported by the Institut Pasteur, the CNRS, the Agence Nationale de la Recherche Laboratoire d'Excellence Revive (Investissement d'Avenir; ANR-10-LABX-73), and by grants from the Fondation pour la Recherche Médicale (Equipe FRM DEQ20120323714 and DEQ20160334881 to PH) and the Fondation ARC pour la Recherche sur le Cancer (to PH). ML was supported by fellowships from the Ligue Nationale contre le Cancer and the Fondation pour la Recherche Médicale. S.H was supported by the Agence Nationale de la Recherche Laboratoire d'Excellence Revive (Investissement d'Avenir; ANR-10-LABX-73). This work was partially funded through ANR-10-INBS-04-06 FranceBioImaging grant to JCOM. The authors acknowledge the Imagopole France—BioImaging infrastructure, supported by the French National Research Agency (Investissement d'Avenir; ANR-10-INSB-04–01) for advice and access to the UltraVIEW VOX system, as well as the Image Analysis Hub of the Imagopole.

# Additional information

## Funding

| Funder | Grant reference number | Author |
|---|---|---|
| Agence Nationale de la Recherche | ANR-10-LABEX-73 | Philippe Herbomel<br>Mylene Lancino<br>Sebastien Herbert |
| Fondation pour la Recherche Médicale | DEQ20120323714 | Philippe Herbomel |
| Fondation ARC pour la Recherche sur le Cancer | | Philippe Herbomel<br>Sara Majello |
| Agence Nationale de la Recherche | ANR-10-INBS-04-06 | Jean-Christophe Olivo-Marin<br>Fabrice De Chaumont |
| Agence Nationale de la Recherche | ANR-10-INSB-04-01 | Jean-Yves Tinevez |
| Ligue Contre le Cancer | | Mylene Lancino |
| Fondation pour la Recherche Médicale | DEQ20160334881 | Philippe Herbomel |
| Fondation pour la Recherche Médicale | | Mylene Lancino |
| Centre National de la Recherche Scientifique | | Anne Schmidt<br>Philippe Herbomel |
| Institut Pasteur | | Jean-Christophe Olivo-Marin<br>Jean-Yves Tinevez |

The funders had no role in study design, data collection and interpretation, or the decision to submit the work for publication.

## Author contributions

Mylene Lancino, Conceived and performed experiments, Analyzed data, Prepared all figures and movies, Wrote figure legends and the Materials and Methods; Sara Majello, Performed experiments; Sebastien Herbert, Developed the code and the methodology for the tracking analysis; Fabrice De Chaumont, Created the 2D algorithm; Jean-Yves Tinevez, Processed images and developed the application of the 2D algorithm; Jean-Christophe Olivo-Marin, Supervised ICY resources; Philippe Herbomel, Conceived the project, Funding and resource acquisition, Analyzed the final data set, Edited the manuscript; Anne Schmidt, Supervised the work, Analyzed the data, Conceptualized, Performed experiments, Wrote and edited the manuscript

## Author ORCIDs

Mylene Lancino (iD) https://orcid.org/0000-0002-5424-9165
Jean-Christophe Olivo-Marin (iD) https://orcid.org/0000-0001-6796-0696
Anne Schmidt (iD) https://orcid.org/0000-0002-8326-0937

## Ethics

Animal experimentation: This study was performed in strict accordance with the recommendations of the regulations of the 2010/63 UE European directives and was supervised by the veterinarian office of the Pasteur Institut (Dr. Myriam Mattei).

## Decision letter and Author response

Decision letter https://doi.org/10.7554/eLife.37355.042
Author response https://doi.org/10.7554/eLife.37355.043

# Additional files

## Supplementary files

• Source code 1. Matlab code for tracking, A-P distances through time and closing speeds.
DOI: https://doi.org/10.7554/eLife.37355.035

• Source code 2. Code of the plugin AortaTracker (used for *Figures 3*, *4* and *5a*-bottom panel, and see Method section).
DOI: https://doi.org/10.7554/eLife.37355.036

• Transparent reporting form
DOI: https://doi.org/10.7554/eLife.37355.037

## Data availability

All data generated or analyzed during this study are included in the manuscript and supporting files. Source data files have been provided for Figures 4 and 5 as well as the Methods. Source code files have been provided for Figures 2, 3 and 4 as well as the Methods.

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
