## [Decision Letter]

Thank you for submitting your article "Anisotropic organization of circumferential actomyosin characterizes hematopoietic stem cells emergence in the zebrafish" for consideration by *eLife*. Your article has been reviewed by Sean Morrison as the Senior Editor, a Reviewing Editor, and two reviewers. The reviewers have opted to remain anonymous.

The reviewers have discussed the reviews with one another and the Reviewing Editor has drafted this decision to help you prepare a revised submission.

Summary:

The work nicely defines imaging of the EHT that is important for stem cell production, and the physical actions that must happen for the budding of stem cells.

Essential revisions:

For the rebuttal, please pay attention to the timing issues addressed in reviewer 1, point number 1. Both reviewers had issues with the myl9b morpholino work, as illustrated by point 2 of reviewer 1 and point 1 of reviewer 2. Blood flow should also be addressed as reviewer 2's point 2. That could be accomplished by the silent heart morpholino or using chemicals.

*Reviewer #1:*

The submitted work by Schmidt et al. builds upon previous work from the Herbomel lab (Kissa and Herbomel, 2010) studying the hematopoietic stem cell (HSC) endothelial to hematopoietic transition (EHT) in zebrafish. The present study uses novel imaging techniques and software analysis to investigate the biomechanics of this process. Based on their data, the authors come to three major conclusions: (i) EHT involves anisotropic contraction along the anterior-posterior (AP) axis facilitated by the heterogeneous organization of contractile circumferential actomyosin, (ii) the contraction is oscillatory with periods of stability that are much longer than other known cell extrusion processes, and (iii) *myl9b* is a target of Runx1 and its amino-terminal phosphorylation site is critical to definitive hematopoiesis. Overall, although the writing could be more clear and succinct and conclusion (ii) could be better supported with requested data and information, this submission provides new and valuable insights into the EHT and seems to be a good fit for publication in *eLife* pending revisions.

1) Conclusion (ii) would be better supported by an improved and more standardized approach to determining length of time of the EHT. This could include more specific and quantitative definitions of the intervals that the authors refer to as "periods" and "pseudo-plateaus." Using time of imaging onset does not appear to be the best initial time from which to measure the duration of the EHT since it is somewhat arbitrary – can this be more standardized from the data already collected? Further, the description for calculating oscillation duration and stabilization was vague, and it was not clear from the text which parameters the authors used to define acceleration/deceleration cycles. And, why were only 7 tracings shown in Figure 1—figure supplement 1C? These tracings don't appear to incorporate the full range of closing patterns observed nor does the 2-4 μm cut-off seem to apply more broadly (e.g., see Figure 2C). Finally, the authors could do a better job comparing (both qualitatively and quantitatively) their findings for EHT to other cell extrusion processes in the Discussion to strengthen their argument for the uniqueness of the EHT and to situate their findings in a broader context.

2) With regard to conclusion (iii), although the authors demonstrate what they claim as a major conclusion, their data leave open a few questions that would be good to address in this work in accordance with published standards for morpholinos (see Stainier et al., 2017). This could include phenocopies with multiple morpholinos or rescue experiments. Could the *myl9a* data (i.e., not regulated by Runx1 but affecting hematopoiesis) be explained by an off-target effect of the morpholino on *myl9b*? What would *myl9b* expression look like with the myl9a morpholinos (potentially add condition to Figure 6—figure supplement 1E)? What does the dose response curve look like and why were such high quantities of morpholinos injected (8-12ng) in the main text when Figure 6—figure supplement 1E suggests that 4ng gave similar changes in mRNA expression levels? Do the myl9aspe2i2 and myl9bspe2i2 morpholinos phenocopy myl9aspi3e4 and myl9bspe3i3, respectively?

*Reviewer #2:*

In this manuscript, the authors described the detail process of endothelial-hematopoietic transition (EHT) including EHT time consumption and occurrence location, hemogenic endothelium morphological change, and dynamic change of intercellular cell junction and cytoskeleton. They regarded EHT as a special pattern of developmental morphogenetic event and cell extrusion. Finally, the authors showed that *myl9b* is a downstream target of Runx1 and plays a role in the dynamic change of cytoskeleton during EHT. Although this is a nice piece of cell biology work, the genetic regulation analysis is relatively rough and weak. It also lacks mechanistic study to explore the signals that trigger the dynamic change of hemogenic endothelium. Addition of these data will significantly enhance the quality of the manuscript.

1) In subsection “Hemodynamic forces control the morphodynamics of the emergence” and Figure 6, the authors argued that Runx1 positively regulates the expression of *myl9b*. Because MO knockdown may have toxicity and off-target effects, the authors should re-examine the expression pattern of *myl9b* in *runx1* mutants rather than *runx1* morphants. More importantly, time-lapse imaging should be performed on *runx1* mutants to find out whether the patterns of Myl9b and F-actin are indeed perturbed in hemogenic endothelium in the mutant fish. Finally, CHIP assay should also be performed to confirm Runx1 indeed binds to the promoter or enhancer of the *myl9b* gene and promotes the expression of *myl9b* in zebrafish.

2) The authors showed that during the EHT, ZO1 and F-actin tend to be enriched in the trijunctions and membrane interfaces that are perpendicular to the blood flow, raising the possibility that blood flow may act as an extracellular mechanical force that triggers or affects the morphological change and cytoskeleton dynamics of HE during EHT. Thus, it would be of great interest to investigate the influence of blood flow on EHT dynamics by manipulating the physical property of blood flow (velocity, volume and viscosity, for example).

---

## [Author Response]

Essential revisions:For the rebuttal, please pay attention to the timing issues addressed in reviewer 1, point number 1.

We have reinvestigated our time-lapse experiments and found 7 EHT cells that were imaged, starting from a flat morphology (hemogenic cells) to their exit (hence including the cup shape stage). We observed that, as is the case for the time-to-exit of cells imaged starting at the cup shape stage, the time intervals between the flat morphology to the cup shape stage are also very variable, with an average of 6.5 hours (ranging from 3.2 hours to 10.33 hours). We have now calculated the apparent complete duration of the EHT, which lasts for more than 10 hours on average (apparent because it is currently impossible to determine the precise starting point at which EHT committed cells, in the hemogenic endothelium, will initiate the emergence). This very long timing of the process reinforces the fact that starting imaging when hemogenic cells committed to perform the EHT are still flat is at risk regarding phototoxicity (which legitimates our initial choice of imaging the process starting at the cup shaped stage). These results are included now in a new version of Figure 1—figure supplement 1, panels A and B (left part), the new Figure 1—video 3 and the related paragraph in the revised main text.

Reviewers had issues with the myl9b morpholino work, as illustrated by point 2 of reviewer 1 and point 1 of reviewer 2.

We have performed 3 major sets of experiments with respect to these concerns, which are outlined in more detail in response to reviewer 1 point 2.

Blood flow should also be addressed as reviewer 2's point 2. That could be accomplished by the silent heart morpholino or using chemicals.

We have performed experiments using a silent heart morpholino and show/illustrate a series of results, with details given in response to reviewer 2’s point 2.

Reviewer #1:

[…] 1) Conclusion (ii) would be better supported by an improved and more standardized approach to determining length of time of the EHT. This could include more specific and quantitative definitions of the intervals that the authors refer to as "periods" and "pseudo-plateaus." Using time of imaging onset does not appear to be the best initial time from which to measure the duration of the EHT since it is somewhat arbitrary – can this be more standardized from the data already collected? Further, the description for calculating oscillation duration and stabilization was vague, and it was not clear from the text which parameters the authors used to define acceleration/deceleration cycles. And, why were only 7 tracings shown in Figure 1—figure supplement 1C? These tracings don't appear to incorporate the full range of closing patterns observed nor does the 2-4 μm cut-off seem to apply more broadly (e.g., see Figure 2C). Finally, the authors could do a better job comparing (both qualitatively and quantitatively) their findings for EHT to other cell extrusion processes in the Discussion to strengthen their argument for the uniqueness of the EHT and to situate their findings in a broader context.

First of all, we wish to thank the reviewer for his/her fruitful recommendations and critics, in particular the ones on (i) the duration of the EHT (and the matching between the temporal evolution of the process and the phases of contraction and stabilization) and (ii) the morpholino work. In addition, and regarding the main text, we believe that things are more clearly explained now and that the text gained very much in quality following the reviewer’s recommendations.

Please find beneath our answers to the specific points.

Regarding the duration of the EHT, we have provided a response to this in the first point in the Essential revisions section of the decision letter.

Regarding the 2-4 μm cut-off, we agree that this does not apply to all cells and have simplified the text accordingly and propose (subsection “EHT dynamics at high spatio-temporal resolution”):

“Secondly, we noticed that the narrowing down of the A-P luminal distance did not occur at a uniform pace, but with phases of contraction and of relative stabilization of variable durations taking place in the limit of an A-P luminal distance of approximately 2 µm, followed by steep decrease until the release of the cell (this fast drop may represent a collapse following the culmination of mechanical tensions). Hence, the heterogeneous duration of the EHT might be explained by these discontinuities, the reasons of which being unknown.”

Regarding the biomechanics of the apical constriction, we have improved the main text so as to describe more clearly the phases of contraction, relative stabilization and oscillations (phases of acceleration/deceleration). As a support to the main text, we have made a new supplemental figure that clearly indicates what we refer to as contraction, stabilization, period and pseudo-plateau (see the new Figure —figure supplement 1 and its legend). The text that we now propose is found in subsection “Intercellular junctions and apical constriction of EHT cells are anisotropic”.

In addition, following the reviewer’s requests, we have improved the quantitative analysis of some of the parameters of the apical contraction and automatized the mathematical analysis of the oscillation periods:

a) For the calculation of the periods, we have automatized the analysis by using a small algorithm based on the prominence (the prominence of a local minimum, or valley, measures how the valley stands out with respect to its depth and location relative to other valleys), see the following link that we have also indicated in a revised paragraph in Materials and methods section, in which one can find also the information for the algorithm): https://fr.mathworks.com/help/matlab/ref/islocalmin.html

b) The calculation of the periods is what is the most important here hence we have not automatized the calculation of the pseudo-plateaus and removed this information from the plot Figure 2D.

Figure 1C shows only 7 tracing because all measurements were performed by hand, which was extremely fastidious (in comparison to the A-P distances measured and reported Figure 2C that were obtained semi-automatically using TrackMate and are more resolutive (because less noisy)). Showing those data in the Supplement file and at that stage of the paper allows introducing, with the support of a morphological criterion, the reason why the EHT process is stochastic and variable in time.

We are astonished to read this comment ‘the authors could do a better job comparing (both qualitatively and quantitatively) their findings for EHT to other cell extrusion processes in the Discussion section’. Indeed, we make quantitative comparisons in the Discussion section, but we focused on few cases (among which 1 extrusion case), the reason being that those have been the most extensively characterized and quantified (ventral cells in *Drosophila* and the ingression of neuroblasts, also in *Drosophila*). In addition, we feel that this would rather be the scope of a review on the topic. We have substantiated with few additional numbers the part regarding the oscillations.

2) With regard to conclusion (iii), although the authors demonstrate what they claim as a major conclusion, their data leave open a few questions that would be good to address in this work in accordance with published standards for morpholinos (see Stainier et al., 2017). This could include phenocopies with multiple morpholinos or rescue experiments. Could the myl9a data (i.e., not regulated by Runx1 but affecting hematopoiesis) be explained by an off-target effect of the morpholino on myl9b? What would myl9b expression look like with the myl9a morpholinos (potentially add condition to Figure 6—figure supplement 1E)? What does the dose response curve look like and why were such high quantities of morpholinos injected (8-12ng) in the main text when Figure 6—figure supplement 1E suggests that 4ng gave similar changes in mRNA expression levels? Do the myl9aspe2i2 and myl9bspe2i2 morpholinos phenocopy myl9aspi3e4 and myl9bspe3i3, respectively?

With regard to the Myl9a and Myl9b morpholinos work, we have performed 3 major sets of experiments with the following outcomes:

We have addressed the potential off-target effect of Myl9a morpholinos on Myl9b mRNAs and vice versa: 3 independent experiments of qRT-PCR were performed addressing the expression levels of each mRNAs upon injection of all different morpholinos independently. In summary, neither Myl9a nor Myl9b morpholinos triggered significant off-target effects. However, with one of the Myl9a morpholinos (spi3e4), we measured an increase in the Myl9b mRNAs indicating that decreasing Myl9a expression can trigger a compensatory response on Myl9b mRNA expression.

These results, more complete, are now substituting the previous results of the previous Figure 6—figure supplement 1, now becoming Figure 8—figure supplement 1.

In addition, because the phenotype triggered by the use of the Myl9a morpholino spi3e4 may be owed to a compensatory response by the upregulation of Myl9b, the left panels of the new Figure 8D and E (previous Figure 6E and F) are now composed with data obtained with the second Myl9a morpholino (spe2i2).

We have phenocopied the phenotype (on the number of CD41:eGFP+ cells in the CHT and sub-aortic space) with a second morpholino for both Myl9a and Myl9b and provide dose-dependences of the effects of morpholinos on the numbers of CD41:eGFP+ cells. The results are now composing part of a new supplemental figure (that is now Figure 8—figure supplement 2A).

We have rescued the sub-aortic and CHT phenotypes triggered by Myl9b (spe3i3) morpholinos (number of CD41:eGFP+ cells in the sub-aortic region and the CHT) by injecting Myl9b encoding mRNAs.

The results are now composing part of the new Figure 8—figure supplement 2B.

Finally, we also addressed particular points in regard to the experiments aimed at interfering with the activity of the Runx1 transcription factor and analysing consequences on Myl9a and Myl9b mRNAs expression (by another approach than using morpholinos).

As the reviewer will see below, we have conducted two other approaches that gave opposite results than the morpholino approach, i.e an increase in Myl9b mRNAs upon decreasing Runx1 expression.

For the entire issue regarding the interference with Runx1, we have prepared 3 figures (Author response image 1, Author response image 2 and Author response image 3).

Author response image 1 recapitulates the Runx1 morpholino work, with the qRT-PCR work (panel B of initial Figure 6), as well as a graph (E) on the dose-dependence of the effects triggered by the morpholino on the number of hematopietic cells in the CHT (with a maximum of approx. 50% decrease in the expression of Runx1 mRNAs at 10ng (D)). This makes the point on why we used such a high concentration (10ng, for the panel B of initial Figure 6).

**Author response image 1. respfig1:** Runx1 morpholino analysis. This figure shows the several experiments that were performed to characterize the runx1 morpholino that was used in this study (first submission). The panel C was originally in the previous Figure 6B. Panels A and B were originally in Figure 6—figure Supplement 1. A and B show the localization of myl9a and myl9b qPCR oligos and amplified products.

(C) qPCR analysis after inhibition of the EHT by injections of runx1 morpholinos. The schematic drawing indicates that only the posterior part of 48 hpf embryos were used for the qPCR (n=2 independent experiments, in triplicates). Relative expression of genes expressed after injection of control or runx1 morpholinos. Error bars represent mean values ± SEM. unpaired t test (****) p<0.0001.

(D) Level of Runx1 mRNA expression after injections of 4, 8 and 10 ng of morpholino. Note that the choice of the 10 ng in the qRT-PCR experiment was because 4 and 8 ng were not sufficient to reach at least 50% reduction in Runx1 mRNAs. This correlates with the data on the phenotype panel E.

(E) Quantification of runx1:nls-mCherry+ cells (Runx1+23 enhancer) in the CHT of 48hpf embryos after 4-12 ng morpholino. Note that the significant reduction in numbers starts at 10ng.

(F) Visualization of the phenotype in embryos at 48hpf; right: CHT..</Author response image 1 title/legend>

Author response image 2 shows the results of the transient CRISPR/Cas9 approach. This makes the point on the decrease of Myl9b mRNA expression when the mutagenesis is performed virtually at saturation.

**Author response image 2. respfig2:** Analysis of Myl9b and Myl9a expression after deletion of part of the Runx1 sequence encoded by exon 4 (Runx1 crispants). This is the first of the 2 strategies that we have used to challenge the phenotypes obtained using the Runx1 morpholino.

(A) schematic representation of the localization of the 2 guides that were used, inside and flanking the 3-prime of exon 4 targeted for CRISPR/Cas9 mutation/deletion.

(B) The table is a list of the different indels obtained after the injection of the couple of guides together with the Cas9. DNA was extracted from 20 embryos heads corresponding to the embryos used to perform the qRT-PCRs in (C). DNA was used as template for PCR, PCR products subsequently migrated on agarose gel, and 2 bands of high and lower MW were obtained. DNA was extracted from both bands, cloned and sequenced. The first 12 lines of the table correspond to clones obtained with the material from the top band and the last line summarizes the 12 clones obtained with the material from the lower MW band and that have the entire expected deletion.

(C) 2 independent qRT-PCR reactions obtained from 2 independent injections. Note that sequencing showed that for the results in the top panel, the mutagenesis was performed virtually at saturation and corresponds to the sequences described in (B). Note that the qRT-PCR results for Runx1 shows the efficiency of exon 4 partial deletion. The amplicons are in the vast majority bearing harmful mutations as shown in panel B. Red bars correspond to mean values ± SD.

*Conclusion: for both experiments myl9b is increased and myl9a is not or slightly increased*. </Author response image 2 title/legend>

Author response image 3 shows the results for the Runx1 deletion mutation approach. This also makes the point on the decrease of Myl9b mRNA expression upon expression of the Runx1 deleted form.

**Author response image 3. respfig3:** Analysis of Myl9b and Myl9a expression after expression of a deleted form of Runx1.

This is the second of the 2 strategies that we have used to challenge the phenotypes obtained using the Runx1 morpholino.

In this strategy, the idea was to inject an mRNA encoding for a deletion mutant of Runx1 (del-Runx1) that contains the runt domain and the flanking putative nuclear localization signal.

(A) Schematic representation of the correspondence between exons and protein domains of Runx1, with the full length form (top) and the deletion mutant (bottom).

(B) Quantification of CD41:eGFP+ cells after injection of 100 pg and 200 pg of del-Runx1 mRNAs in the AGM (left:top) and the CHT (left:bottom). Right panels, visualization of the phenotypes in AGM and CHT. This result validated partially the approach of the interference upon expression of del-Runx1.

(C) 2 independent qRT-PCR reactions obtained from 2 independent injections at 35hpf (top panel) and 48hpf (bottom panel). Note that the absence of increase in Runx1 level at 35 hpf indicates that the mRNA encoding for del-Runx1 was probably already degraded and that the decrease at 48 hpf may indicate that del-Runx1 destabilizes the endogenous mRNA. Red bars correspond to mean values ± SD.

*Conclusion: for both experiments myl9b is increased and myl9a is not or slightly increased*.</Author response image 3 title/legend>

For interfering with the activity of Runx1 and analysing consequences on Myl9a and Myl9b mRNAs expression, we used 2 approaches (i) transient CRISPR/Cas9 (generation of crispants, see (Burger et al., 2016)), (ii) expression of a truncated form of Runx1 deleted of the transactivation domain (supposedly a dominant-negative form, as characterized previously (Tracey et al., 1998)).

Here are the experiments that we have performed (2 sets of experiments):

To obtain the crispants, we used 1 couple of guides hybridizing in or downstream the sequence of exon 4 of the zebrafish runx1 gene (the exon that encodes for the most distal part of the runt domain that contains the essential amino acid residues involved in DNA binding). 2 independent injections were performed (each with 2 independent qRT-PCR analysis including measurements of Runx1 exon 4 deletion efficiency, expression levels of Myb (internal control), Myl9a and Myl9b mRNAs), with 1 experiment performed virtually at saturation.

In these conditions, we measured reproducibly and by qRT-PCR, small decrease for the positive control (Myb), no to little variation in Myl9a mRNAs, and a reproducible increase in Myl9b mRNAs.

We designed a potential dominant negative form of Runx1 (that is deleted of the transactivation domain and remains with the runt-DNA and CBFβ interaction domain and, in addition, a flanking downstream sequence that encompasses a putative nuclear targeting signal, i.e a mutant form that retains amino-acids 1-238. A similar deletion mutant (but lacking the putative nuclear targeting signal) was used and characterized as a dominant negative form in *Xenopus* (see Tracey et al., 1998).

Injection of mRNAs encoding for this mutant form triggered quite significant decrease of CD41:eGFP+ hematopoietic cells in the CHT at 48hpf (in addition to small haemorrhaging within the brain and, apparently, the pericardial space, which appears to be consistent with the phenotype obtained with Runx1 deficient mouse embryos (Okuda et al., 1996; Wang et al., 1996)). This validated the approach and encouraged us investigating further the effect of the Runx1 deletion mutant on the expression levels of Myl9a, Myl9b and Myb (the control). We found, for Myl9a and Myl9b approximately the same results as for the crispants: no to little variation in Myl9a mRNAs, a reproducible increase in Myl9b mRNAs, but no decrease in the positive control (Myb).

In conclusion, these 2 sets of data are contradicting the results obtained by the Runx1 morpholino approach, i.e. a potential involvement of Runx1 in controlling positively the expression of Myl9b. A possible explanation for the discrepancies, outside from toxic and/or off-target effects of morpholinos, is the activation of compensatory responses upon strong modification of Runx1 expression level that affect the consistency of results as well as the sensitivity of hemogenic cells to Runx1 dosage.

Because of these inconsistencies, we removed the results obtained with the Runx1 morpholino (which was previous Figure 6, panel B). We believe that this does not impairs the impact of this work that is now very much substantiated by the outcome of the new investigations that we have conducted on the influence of blood flow on the EHT dynamics.

Reviewer #2:

[…] 1) In subsection “Myosin regulatory light chains 9a and 9b are required for definitive hematopoiesise” and Figure 6, the authors argued that Runx1 positively regulates the expression of myl9b. Because MO knockdown may have toxicity and off-target effects, the authors should re-examine the expression pattern of myl9b in runx1 mutants rather than runx1 morphants. More importantly, time-lapse imaging should be performed on runx1 mutants to find out whether the patterns of Myl9b and F-actin are indeed perturbed in hemogenic endothelium in the mutant fish. Finally, CHIP assay should also performed to confirm Runx1 indeed binds to the promoter or enhancer of the myl9b gene and promotes the expression of myl9b in zebrafish.

First of all, we wish to thank the reviewer for his/her very fruitful recommendation on investigating the influence of blood flow on the EHT dynamics. As the reviewer will see, the outcome of the experiments is very interesting and substantiates very much the idea that blood flow has an influence on several aspects of the EHT biomechanics. In addition, the comments on the morpholinos work has been very valuable also and led to substantial revision of our work. Please find beneath our answers to the specific points.

We agree with the reviewer that we have not shown that the *myl9b* gene is a direct target of Runx1 (which we wrote was a possibility since it was shown to be the case for mouse cells). This would require, as suggested by the reviewer, a CHIP assay. The CHIP assay is the only experiment that we have not performed because we do not handle this technique in the laboratory and, taking information from other laboratories on our campus, this kind of experiment cannot reasonably be conducted in 2 months (the time we were given with this revision for which virtually all points raised by the reviewers were addressed). Importantly, and as the reviewer will see beneath, since the new experiments that were performed during this revision in order to interfere with the expression of Runx1 (using CRISPR/Cas9 and the expression of a deleted form of Runx1) gave inconsistent results with the morpholinos, we have removed the qRT-PCR data on Runx1/Myl9b, Myl9a from the paper. We believe that this does not harm the overall outcome of our work that is now more substantiated on its main message, i.e the importance of mechanical forces of the blood flow in the biomechanics of the EHT, which is a new part of the revised version and which arose after the very constructive suggestion of the reviewer.

We addressed particular points in regard to the experiments aimed at interfering with the activity of the Runx1 transcription factor and analysing consequences on Myl9a and Myl9b mRNAs expression (by another approach than using morpholinos). This information is given in response to reviewer 1’s second point.

For the comment “time-lapse imaging should be performed on runx1 mutants to find out whether the patterns of Myl9b and F-actin are indeed perturbed in hemogenic endothelium in the mutant fish”:

To perform a pattern analysis at a good resolution, we need to use ideally our 2D-map treatment of images, which we cannot perform presently because, as we mention in the text of our paper, we do not have enough sensitivity with our fish lines, neither for Myl9a, nor for Myl9b. The experiments requested by the Reviewer are extremely difficult experiment to perform (in particular in the 2-3 months of a revision) owing to low sensitivity (in particular for Myl9a and b) and variability (in particular when imaging stochastic processes, as EHT is). Hence, this type of experiments would need, at least, to have at hand a well-characterized stable fish line mutant for Runx1, which we presently do not have. This would indeed require such a line because, for Myl9b (and so would be the case for Myl9a), we would have to inject a Myl9b encoding plasmid (to increase the chance to have enough sensitivity of detection but having to face the problem of mosaicism in transient transgenesis). We currently do not handle transient transgenesis for expression of Lifeact and we can anticipate on the same type of difficulties.

So, we feel that we are not so far from addressing the reviewer’s request (although rather indirectly) by showing the alteration of the F-actin pattern after the injection of the sih morpholino (see the new Figure 6, bottom of panel B), given that the blood flow controls the expression of Runx1.

2) The authors showed that during the EHT, ZO1 and F-actin tend to be enriched in the trijunctions and membrane interfaces that are perpendicular to the blood flow, raising the possibility that blood flow may act as an extracellular mechanical force that triggers or affects the morphological change and cytoskeleton dynamics of HE during EHT. Thus, it would be of great interest to investigate the influence of blood flow on EHT dynamics by manipulating the physical property of blood flow (velocity, volume and viscosity, for example).

We have performed experiments using a silent heart morpholino and show/illustrate a series of results that are listed below. It should be emphasized that lack of blood flow triggered a quite significant decrease in the diameter of the aorta and distortions of its wall rendering the imaging more difficult than in normal conditions. However, we have been able to obtain few time-lapse sequences with interesting results regarding the orientation of the emergence and the morphodynamics of emerging cells (see below). We also explored further the phenotype, in particular with our 2D-algorithm to analyse the localization/organization of F-actin and ZO1. Impairing blood flow also dramatically affected the results of image treatment with the 2D-algorithm because these distortions project the signals outside from the crown drawn on aortic sections and supposed to contain signals of highest intensities (which are the signals projected on the 2D plane). Hence, we have narrowed down the images obtained by 2D-mapping to the plane of the aortic floor (that was less distorted than the aortic roof that flattened a lot, see Figure 6A) and obtained interesting information on the organization of F-actin (see below).

We also investigated the consequence of impairing blood flow on the localization/organization of ZO1. Unfortunately, fluorescence signals were too low and aorta too distorted to perform 2D-mapping (in addition, we wish to inform the reviewer that we had a serious incident in our animal facility and lost several of our fish lines, including the UAS:eGFP-ZO1 line that had been highly selected over several generations to reduce mosaicism that was very strong in the dorsal aorta. Hence, the experiments that were performed here with the sih morpholino were on transient expression, which is why we had very little amounts of cells expressing eGFP-ZO1 in the aortic floor (see new Figure 6, panel C)). However, although we were not able to perform 2D-mapping with our Z-stacks obtained with fishes transiently expressing eGFPZO1, 3D-projections and 3D-analysis with the Imaris software were very helpful and revealed alterations of eGFP-ZO1 localization in cells of the hemogenic endothelium, with altered recruitment at the sub-plasmalemmal level and apparent increase of the cytoplasmic pool. In addition to the morphological alterations of the dorsal aorta, we show/illustrate: (a) the absence, on the ventral side of the dorsal aorta, of EHT cells with their luminal/apical membrane oriented towards the sub-aortic space (in the same direction than the basal membrane) but, rather, the presence of ovoid cells (with the luminal/apical membrane protruding into the aorta lumen). 14 embryos were imaged all along the dorsal aorta at 48 hpf and 3 additional ones were used for time lapse sequences that lasted for 6 to 12 hours; in none of these embryos did we see any cup shaped EHT undergoing cells; (b) the emergence of ovoid cells in the lumen of the aorta (which we never saw before); (c) the clear modification in the organization of both F-actin and ZO1 in hemogenic and EHT cells. (d) the maintenance of mitosis of cells embedded in the aortic floor (most probably in the hemogenic endothelium). (e) evidence for the death of some of the cells that have emerged in the sub-aortic space (in the time window of our time-lapse experiments).

In addition, we also interfered with the heart beating by using treatment with tricaine. In these conditions, the inward bending of the apical/luminal membrane is abrogated and cells performing the emergence towards the sub-aortic space maintained the direction of the emergence towards the sub-aortic space (we introduced that information in the text as Data not shown).

Overall, these results show that hemodynamic forces contribute to the bending of the apical/luminal membrane of EHT cells towards the sub-aortic space (the cupshaped cells) as well as the direction of the emergence (the sub-aortic space) and reinforce the peculiarity of the EHT biomechanics, influenced by – and adapted to – the mechanical tensions exerted by the blood flow (Abstract).

These new results are included in a new figure, Figure 6, and 3 new Movies (Figure 6—video 1 (which is not a movie sensu stricto but a series of Z stacks illustrating the aortic morphology, intra-aortic cells, and the presence of ovoid cells included in the aortic floor), Figure 6—video 2 (illustrating the dynamics of the EHT emergence without flow) and Figure 6—video 3 (illustrating, by 3D-projections and 3D-analysis with the Imaris software, the alterations of eGFP-ZO1 localization in cells of the hemogenic endothelium). These results are contained in subsection “Hemodynamic forces control the topology and the dynamics of the emergence” and led to some modifications of the text on the sequential steps of the EHT (see subsection “EHT sequential steps and key features”).